RESEARCH

# Gut microbiota DPP4-like enzymes are increased in type-2 diabetes and contribute to incretin inactivation

Marta Olivares[1], Paula Hernández-Calderón[2], Sonia Cárdenas-Brito[2], Rebeca Liébana-García[1], Yolanda Sanz[1*] and Alfonso Benítez-Páez[1,2*]

*Correspondence:
yolsanz@iata.csic.es;
abenitez@iata.csic.es

[1] Institute of Agrochemistry and Food Technology, Microbiome, Nutrition and Health Research Unit, Spanish National Research Council, IATA-CSIC, 46980 Paterna-Valencia, Spain
[2] Principe Felipe Research Center (CIPF), Host-Microbe Interactions in Metabolic Health Laboratory, 46012 Valencia, Spain

## Abstract

**Background:** The gut microbiota controls broad aspects of human metabolism and feeding behavior, but the basis for this control remains largely unclear. Given the key role of human dipeptidyl peptidase 4 (DPP4) in host metabolism, we investigate whether microbiota DPP4-like counterparts perform the same function.

**Results:** We identify novel functional homologs of human DPP4 in several bacterial species inhabiting the human gut, and specific associations between *Parabacteroides* and *Porphyromonas* DPP4-like genes and type 2 diabetes (T2D). We also find that the DPP4-like enzyme from the gut symbiont *Parabacteroides merdae* mimics the proteolytic activity of the human enzyme on peptide YY, neuropeptide Y, gastric inhibitory polypeptide (GIP), and glucagon-like peptide 1 (GLP-1) hormones in vitro. Importantly, administration of *E. coli* overexpressing the *P. merdae* DPP4-like enzyme to lipopolysaccharide-treated mice with impaired gut barrier function reduces active GIP and GLP-1 levels, which is attributed to increased DPP4 activity in the portal circulation and the cecal content. Finally, we observe that linagliptin, saxagliptin, sitagliptin, and vildagliptin, antidiabetic drugs with DPP4 inhibitory activity, differentially inhibit the activity of the DPP4-like enzyme from *P. merdae*.

**Conclusions:** Our findings confirm that proteolytic enzymes produced by the gut microbiota are likely to contribute to the glucose metabolic dysfunction that underlies T2D by inactivating incretins, which might inspire the development of improved antidiabetic therapies.

**Keywords:** DPP4 enzyme, GLP-1, Gliptins, Incretins, Gut microbiome, Obesity, Glucose metabolism, Type 2 diabetes

## Background

The causal relationship between obesity and comorbidities such as type 2 diabetes (T2D) and the gut microbiota has long been established; however, our understanding of the specific mechanisms driving this relationship remains limited [1]. Microbial mimicry

of host activities has been proposed as a mechanism by which pathogens manipulate host cellular functions [2], and recent evidence indicates that mimicry is also a strategy that can be used by commensal bacteria in the gut. For example, *Bacteroides vulgatus* produces the metabolite N-acyl-3-hydroxypalmitoyl-glycine, which can structurally mimic long-chain N-acyl amides that function as mammalian signaling molecules. Both metabolites activate the GPR132/G2A receptor, which has been implicated in autoimmune disease and atherosclerosis [3, 4]. Similarly, the caseinolytic protease ClpB from the gut commensal *Eschericia coli* has been shown to translocate and mimic the action of α-melanocyte-stimulating hormone to control host satiety [5, 6].

Dipeptidyl peptidase 4 (DPP4 or CD26) is a serine protease that rapidly inactivates the incretin peptides glucagon-like peptide-1 (GLP-1) and gastric inhibitory polypeptide (GIP) to modulate postprandial insulin secretion and glycemia [7]. Accordingly, administration of DPP4 inhibitors is a pharmacological strategy to control T2D by prolonging the half-life of GLP-1 and GIP [8]. Plasma DPP4 activity increases with obesity, which has been postulated to contribute to reduced incretin activity in the setting of obesity and insulin resistance [9, 10]. However, the generation of mice with loss of DPP4 activity in multiple tissues and cell types has shown that not all cellular sources of DPP4 activity affect glucose metabolism [11, 12]. For example, endothelial DPP4 cleaves inactive GLP-1 and GIP [11], but adipocyte- and hepatocyte-derived DPP4 are dispensable for glucose control [12]. We recently demonstrated the ability of the gut microbiota to produce DPP4-like activity in germ-free and conventionally colonized mice, leading us to propose that gut microbes are potential mediators of incretin inactivation [13]. Subsequent studies have identified the primary sources of human DPP4 (hsDPP4) homologs in species of the genera *Bacteroides*, *Parabacteroides*, and *Porphyromonas* [14–16]. In the case of *Bacteroides* species, it has been reported that this gene is important for envelope integrity and that DPP4 loss reduces bacterial fitness during in vitro growth within a diverse community [14], but not when growing alone [16]. In addition, a peptidase activity complementary to that of DPP4, termed Xaa-Pro dipeptidyl-peptidase (PepX), has been identified in lactic acid bacteria commonly found in fermented foods and the human gut, such as *Lactobacillus* spp., *Lactococcus* spp., and *Streptococcus* spp. [17, 18].

In line with our proposition that gut microbial DPP4 activity might influence host metabolism [13], some species of *Bacteroides* have recently been reported to inactivate GLP-1 in mice fed a high-fat diet [16]. Here, we investigated how bacterial DPP4, which is prevalent in patients with T2D, impairs gut hormone function and evades the catalytic inhibition of drug therapies for T2D. This study represents an advanced over recent microbiome research in this field [16] by identifying a novel gut bacterial species that harbor functional hsDPP4 homologs and associating the bacterial DPP4 genes with T2D and glycemia using metagenomic data from a large cohort of patients [19]. We also present an in-depth protein characterization and demonstrate the potential of the bacterial DPP4 to be secreted extracellularly and not only hydrolyze human incretins (GLP-1 and GIP) but also human neuropeptides (peptide tyrosine tyrosine—PYY and neuropeptide Y—NPY), and the incretin functions are confirmed in a preclinical model. Finally, in an expanded microbiota-drug interaction assessment, we demonstrate the differential interaction between bacterial DPP4 and four prescribed anti-diabetic drugs with DPP4 inhibitory activity. Our results shed light on the molecular mechanisms by

which specific gut bacterial species alter the host glucose metabolism and the efficacy of DPP4 inhibitors, providing new opportunities to improve drug development for the treatment of T2D.

## Results

### DPP4-like genes are widespread across human gut *bacteria*

We used a bottom-up approach to search for bacterial DPP4-like genes in the metagenomes of human subjects with metabolic disorders [20], where the presence of the enzymes may have clinical implications. We retrieved DPP4-like genes encoded by different Gram-negative bacteria belonging not only to the genus *Bacteroides*, but also to the genera *Alistipes*, formerly *Prevotella*, *Barnesiella*, and *Paraprevotella*. We constructed a phylogenetic tree using amino acid sequence information from more than 1600 non-redundant bacterial peptidases, including PepX (belonging to the so-called S15 family) and DPP4-like and eukaryotic proteins (family S9). We included the PepX family because it is a family of peptidases exerting similar catalytic functions [21, 22], often confused with DPP4. This analysis reinforces the evidence of their independent lineage and disparate phylogenetic distribution. After applying advanced phylogenetic methods, we obtained a maximum likelihood tree explaining the evolutionary relationship between the two peptidase families (Fig. 1A). According to Akaike's Information Criterion (AIC), and among the 120 models tested, the WAG + I + G model (Whelan and Goldman model + invariable amino acid sites + gamma distribution over substitution rates) of protein evolution better explained the relationship between the S15 and S9 peptidase families and determined both as independent lineages. Notably, PepX proteins were found exclusively in Gram-positive bacteria whereas DPP4-like proteins were found exclusively in Gram-negative bacteria. Taxonomic assessment revealed that PepX peptidases are predominantly found in the families Streptococcaceae and Lactobacillaceae, and their respective genera, within the Bacillota phylum (formerly Firmicutes) (Fig. 1A). The families Enterococcaceae, Leuconostocaceae, and Sporolactobacillaceae were also represented to a minor extent. Contrastingly, DPP4-like proteins were widely and exclusively distributed across several families of the Bacteroidota phylum (formerly Bacteroidetes) and were evident in several species from the Flavobacteriaceae, Prevotellaceae, Porphyromonadaceae, and Bacteroidaceae families (Fig. 1A). Furthermore, the DPP4-like family proteins showed greater diversity and genetic distances between members, and proteins from Flavobacteriaceae members appeared to have evolved independently of other Bacteroidota families. Overall, the amino acid sequences and the evolutionary relationships between the S15 and S9 peptidase families support a process of functional convergence [23]. Moreover, we found proteins corresponding to the three eukaryote proteins (humans, rat and mouse), with identity values of 25–35% (within the *twilight zone* of sequence similarity) (Fig. 1B). The position of the eukaryotic DPP4 enzymes in the tree seem to not support a phylogenetic relationship with bacterial DPP4-like enzymes, suggesting that they did not diversify from a common ancestor. Indeed, the closest relationship between human, rat, and mouse DPP4 enzyme sequences was established with DPP4-like sequences from environmental bacterial species of the Balneolaceae family (genera *Gracilimonas* and *Aliifondinibius*), which are marine species with a high frequency of horizontal gene transfer [24].

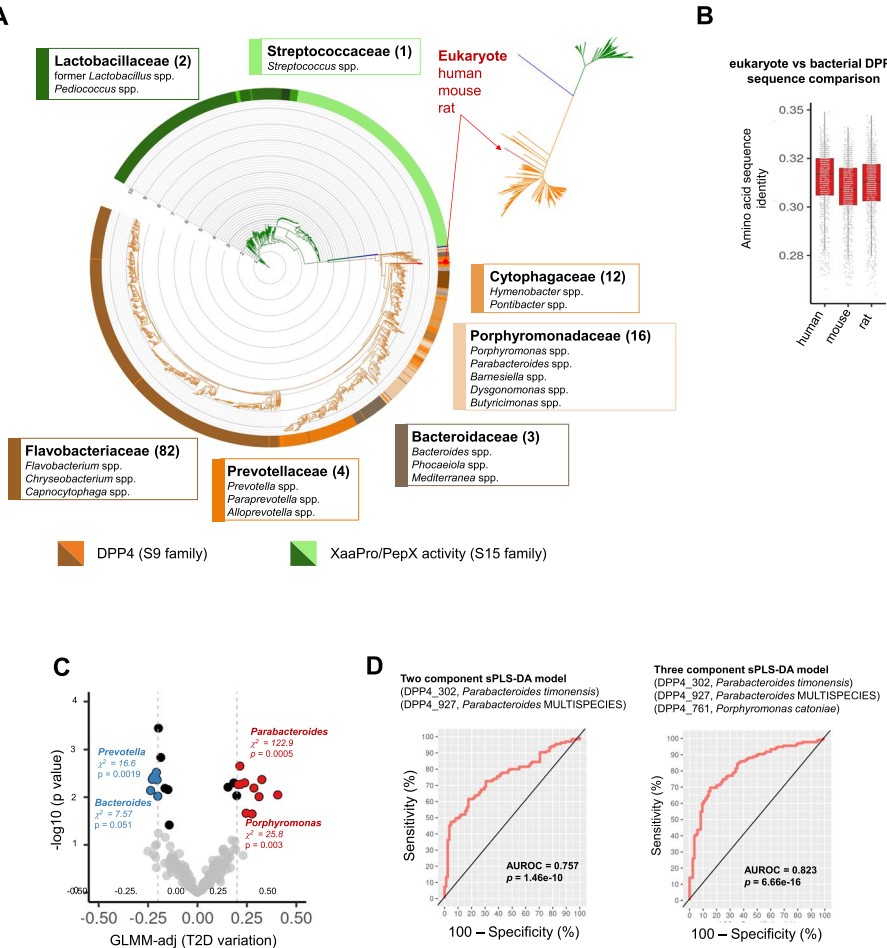

**Fig. 1** **A** Taxonomic distribution and phylogenetic analysis of PepX and DPP4 peptidases. The phylogenetic relationship between both families of bacterial peptidases is illustrated by a circular tree. A small unrooted tree is also shown at the top-right corner. The tree includes the bait used in the PSI-Blast search for distant homologs and the human, mouse, and rat sequences of DPP4 enzymes for comparison. The outlier consisted of an S17 peptidase sequence from *Dichelobacter nodosus* (SwissProt ID P19577). The annotation and grouping of protein sequences at the family and genus level was performed through the inspection of species retrieved from PSI-Blast search across the hierarchical levels of SILVA (release 138) [25]. Branch colors are according to the peptidase family grouping: green for XaaPro peptidases (S15), orange for DPP4 peptidases (S9), and blue for the outlier (S17). Strip colors indicate the taxonomic distribution of sequences accordingly. Labels for the most abundant taxonomy groups in terms of families and genera (numbers within parenthesis) are shown. The tree was visualized using the iTOL web server [26]. **B** Distribution of amino acid identities of eukaryote proteins *versus* bacterial DPP4-like sequences is shown as a boxplot. **C** Metagenome abundance of DPP4-like proteins for classification of type 2 diabetes (T2D). A volcano plot shows the abundance variation of bacterial DPP4-like proteins in human gut metagenomes linked to T2D. Generalized linear mixed models (GLMM) estimates and *p*-values obtained after controlling for covariates (sex, age, body mass index) were used to identify proteins associated with T2D ($N = 135$) *versus* no-T2D controls ($N = 85$). A $2 \times 2$ contingency table-based *chi*-squared test with Monte-Carlo simulation was computed to determine protein enrichment at the genera level. The threshold for selection was set at $> 0.20 \, || < -0.20$ (light grey lines). Red points indicate an association with T2D samples, and blue ones with no-T2D samples. Enrichment genera categories are shown as insets in the main plot. **D** Area under the receiver operating characteristic (AUROC) curve resulting from the sparse partial least square discriminant analysis (sPLS-DA) to determine the performance of the DPP4-like proteins to classify T2D samples. Two models with two and three components (informative genetic traits) support the higher discriminant power of DPP4-like gene abundance from *Parabacteroides* and *Porphyromonas* species (on top) to discern T2D samples. AUROC and *p*-values are shown as insets in respective subpanels. Standard errors (SE) on AUC curves were calculated with the *auctestr::se_auc* R function retrieving 0.032 and 0.027, respectively

Analysis of the protein architecture of the bacterial DPP4-like proteins revealed a leader peptide similar to that of the human counterpart (Additional file 1: Figure S1A-C), which is involved in transmembrane translocation, suggesting extracellular secretion. Furthermore, the amino acid profile of all the analyzed DPP4-like proteins (bacterial and eukaryote, $N = 971$) indicated that all proteins contain the same catalytic motif of S9 serine proteases (GWSYGGY) according to the MEROPS database (Additional file 1: Figure S1D) [22].

### Increased abundance of microbial DPP4-like genes is associated with T2D

To investigate the association between microbial DPP4 and T2D, we evaluated the abundance of all retrieved DPP4-like bacterial genes in the metagenome of 220 subjects with T2D from a previous study [19]. A GLMM (generalized linear mixed models) analysis controlling for covariates indicated that some DPP4-like genes were more abundant in patients with T2D than in non-T2D controls (Fig. 1C). The largest differences were detected for DPP4-like genes from *Parabacteroides* sp. (DPP4_927, GLMM-adj $= 0.40$, *p*-adj $= 0.009$), *Chryseobacterium* sp. YR203 (DPP4_609, GLMM-adj $= 0.33$, *p*-adj $< 0.001$), *Parabacteroides merdae* (DPP4_931, GLMM-adj $= 0.31$, *p*-adj $< 0.001$), *P. timonensis* (DPP4_302, GLMM-adj $= 0.29$, *p*-adj $= 0.006$), and *Parabacteroides* sp. SN4 (DPP4_249, GLMM-adj $= 0.27$, *p*-adj $= 0.023$). On average, *Parabacteroides* DPP4-like genes are 1.28-fold more abundant in T2D patients than in controls. The most remarkable difference is found in the DPP4_927 gene, which shows a 1.98-fold greater relative abundance in T2D patients compared to controls (0.064% vs 0.033%, median, respectively). A genus-level (taxa grouped) enrichment approach revealed that DPP4-like genes are mainly encoded by species of two genera in patients with T2D: *Parabacteroides* (*chi-squared* $= 122.9$, $p = 5.0e-4$) and *Porphyromonas* (*chi-squared* $= 25.8$, $p = 0.003$). Contrastingly, DPP4-like genes encoded by species of the genera *Bacteroides* and *Prevotella* were less abundant in patients with T2D (Fig. 1C). We then used a machine learning approach (sparse Partial Least Square Discriminant Analysis—sPLS-DA) to question whether the DPP4-like genes could be used as a classification trait in T2D. Robust models were built using two and three more informative features from the DPP4-like dataset, which revealed that the genes from *Parabacteroides timonensis* (DPP4_302), *Parabacteroides* MULTI-SPECIES 99% (annotated as consensus sequence of several *Parabacteroides* species at 99% sequence identity) (DPP4_927), and *Porphyromonas catoniae* (DPP4_761) had high discriminatory power when evaluated in combination (AUROC $= 0.823$, $p = 6.7e-16$) (Fig. 1D).

### Bacterial DPP4 is secreted like its human counterpart

To understand the extent to which the function of bacterial DPP4-like proteins is similar to hsDPP4 and to provide particular insights into the function of this divergent protein from those previously characterized [14, 16] (sharing ~ 52% sequence identity with those of *Bacteroides* species), we cloned the DPP4 gene from *P. merdae*, the member of the gut microbiome that is more strongly associated with T2D and the *Parabacteroides* species that is more abundant in the human gut [27]. We cloned the full-length DPP4-like gene (pmDPP4) and a truncated version lacking the predicted leader sequence of 23 amino acids at the N-terminus (pmDPP4Δ23) into a His-tag expression vector, and confirmed

over-expression of the recombinant proteins in *E. coli* BL21-DE3 cells (Additional file 2: Figure S2A). Mass spectrometry analysis revealed that cell-free supernatants from *E. coli cultures* expressing pmDPP4, but not those expressing pmDPP4Δ23, could cleave GLP-1, as reflected by the expected shift in the molecular mass of GLP-1 from 3354 to 3147 Da resulting from the cleavage of the His-Ala N-terminal dipeptide (Fig. 2A–C). We next investigated the subcellular localization of pmDPP4 and pmDPP4Δ23 proteins in bacterial cultures using anti-His immunogold staining and transmission electron microscopy. Results showed that full-length pmDPP4 tended to localize to the inner membrane and was also secreted and embedded in the extracellular polysaccharide capsule [28], whereas pmDPP4Δ23 tended to localize in the cytosol (Fig. 2D–F).

To confirm the role of the pmDPP4 signal peptide in directing secretion and allowing extracellular GLP-1 hydrolysis, we performed an in vitro enzymatic assay using the synthetic dipeptide Gly-Pro p-nitroanilide (gpPNA) as a substrate. We also measured the rate of cell lysis by titrating lactate dehydrogenase (LDH) activity during the preparation of cell-free supernatants, to control for spontaneous DPP4-like activity. We were unable

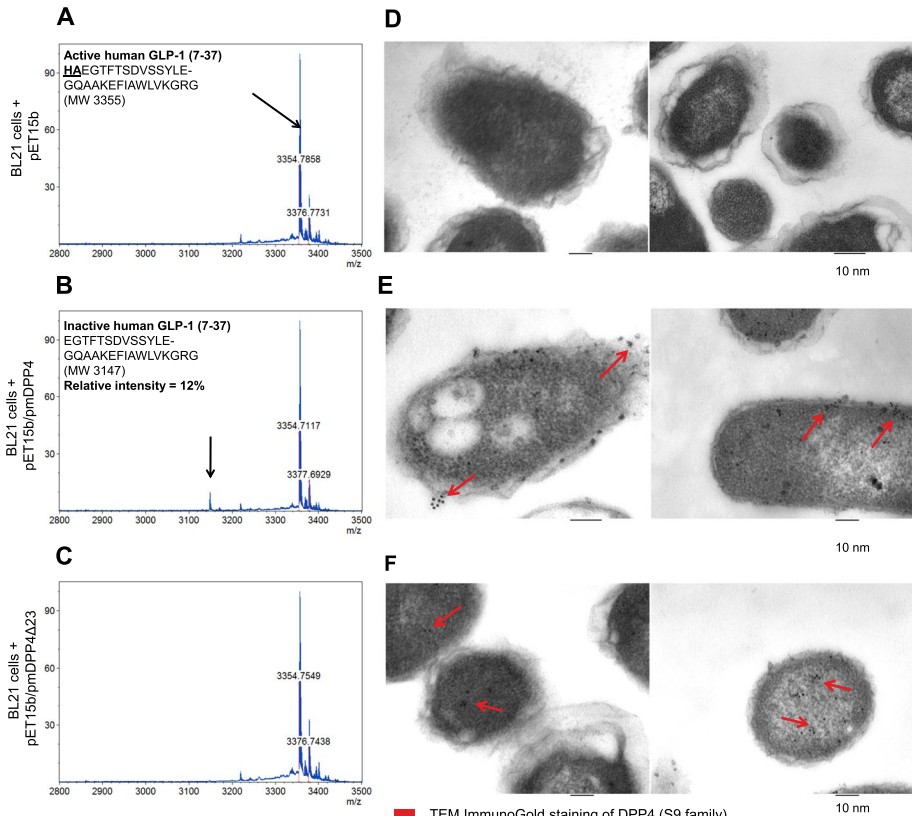

**Fig. 2** Secretory signal detection in the pmDPP4 protein. MALDI-TOF mass spectrometry monitoring of human GLP-1 incretin inactivation in enzymatic reactions using cell-free supernatants from *E. coli* BL21 cells carrying the pET15b (**A**), pET15b/pmDPP4 (**B**), and pET15b/pmDPP4Δ23 (**C**) constructs upon induction. Active form of GLP-1 (MW 3,355) was detected in all cell-mediated hydrolysis assays and the expected inactive form (MW 3,147) resulting from N-terminal dipeptide cleavage was only evident in cells carrying pET15b/pmDPP4 plasmid encoding full-length protein. Anti-His transmission electron microscopy immunogold staining of *E. coli* BL21 cells carrying the pET15b (**D**), pET15b/pmDPP4 (**E**), and pET15b/pmDPP4Δ23 (**F**) constructs upon induction. Red arrows indicate predominant distribution of 6 × His-tagged proteins for **E** and **F** panels. Scale bars at 100 nm (bars below TEM images). MW: monoisotopic mass

to detect bacterial cell lysis during the reaction, and thus the increase in DPP4-like activity observed in *E. coli* BL21-DE3 cells expressing full-length pmDPP4 (but not in cells expressing pmDPP4Δ23) was likely mediated by secretion of the accumulated enzyme into the extracellular medium (Additional file 2: Figure S2B). Thus, the leader peptide, which is evolutionarily conserved in bacterial DPP4-like proteins, is likely required for protein secretion and favors interaction with their substrates.

### Parabacteroides merdae DPP4-like enzyme hydrolyzes human incretins and neuropeptides in vitro

As our data suggested that the pmDPP4Δ23 protein was likely to be the mature form produced and secreted by *P. merdae*, we tested additional functional aspects of the protein. Gel filtration analysis indicated that pmDPP4Δ23 dimerizes in solution to a protein of ~130 kDa, similar to the quaternary structure of the human counterpart (Additional file 2: Figure S2C) [29]. To assess whether the pmDPP4Δ23 protein was similarly active to the full-length protein produced and secreted by *E. coli* BL21-DE3 cells, we determined its hydrolytic activity (Additional file 2: Figure S2C,D). As a qualitative approach, we tested whether pmDPP4Δ23 could cleave the incretin and neuropeptide targets of hsDPP4 in a comparative assay with hsDPP4 by MALDI-TOF–MS (Additional file 2: Figure S2D). Under equimolar conditions and equal substrate concentrations, the pmDPP4Δ23 protein cleaved GLP-1, GIP, PYY, and NPY peptides as efficiently (and identically) as hsDPP4 (100% substrate hydrolysis) (Fig. 3). Specifically, GLP-1 and GIP peptides were completely processed by pmDPP4Δ23 at the first two N-terminal amino acids (His-Ala and Tyr-Ala, respectively), consistent with the molecular mass shift (Fig. 3A,B). pmDPP4Δ23 also hydrolyzed the anorexigenic hormone PYY and the neuropeptide NPY by removing the Tyr-Pro N-terminal dipeptide in both cases, as for hsDPP4 (Fig. 3C,D).

### Parabacteroides merdae DPP4-like enzyme inactivates GLP-1 and GIP in vivo

To investigate whether the DPP4-like enzyme cleaves and inactivates incretins in vivo, we administered mice with *E. coli* BL21-DE3 carrying the plasmid encoding full-length pmDPP4 or the empty vector. To mimic obesity-associated low-grade inflammation and gut barrier defects, mice were injected intraperitoneally with lipopolysaccharide (LPS) to stimulate metabolic endotoxemia, which would facilitate the translocation of bacterial products from the intestinal lumen (Fig. 4A). We observed significantly higher DPP4 activity in the cecal content and portal vein of the group receiving *E. coli* carrying the bacterial DPP4 (Fig. 4B,C). Supporting our hypothesis, the increase in DPP4 activity in the portal circulation was associated with a significant reduction in the active forms of

(See figure on next page.)
**Fig. 3** In vitro function of the pmDPP4 protein. MALDI-TOF mass spectrometry spectra of hydrolytic reactions performed in the presence of pmDPP4Δ23 or hsDPP4 proteins. Panels are distributed in terms of substrate used and the enzyme added (**A–D**). The substrates used for reactions are depicted on top of every panel series with the respective sequence information and the theoretical molecular weight (MW, monoisotopic mass). Delta molecular mass values were calculated from the monoisotopic masses of products observed in the control hsDPP4 reactions. The *x*-axis indicates the molecular mass values in terms of m/z, and the *y*-axis indicates relative intensity (R.I.) of the most abundant peptide species

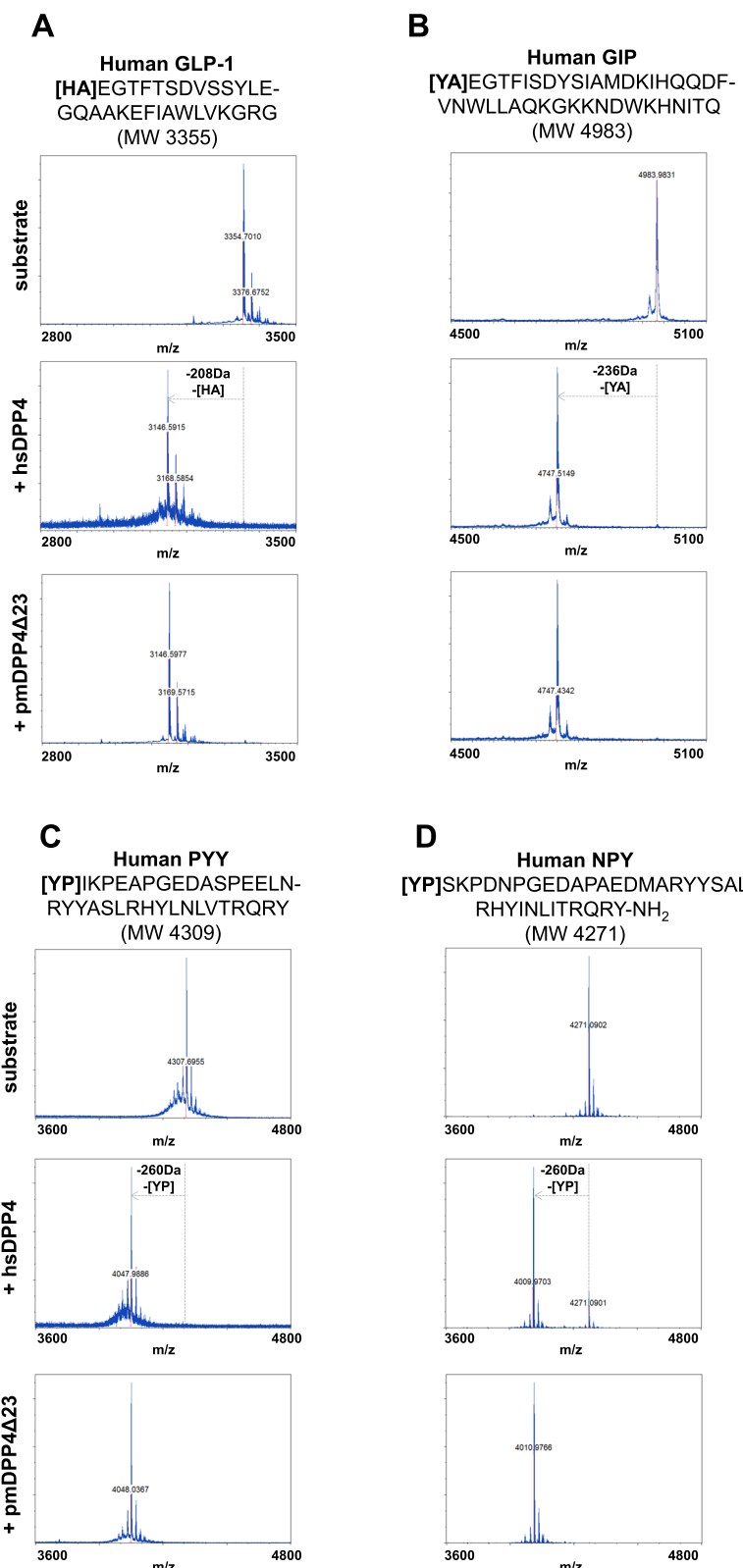

**Fig. 3** (See legend on previous page.)

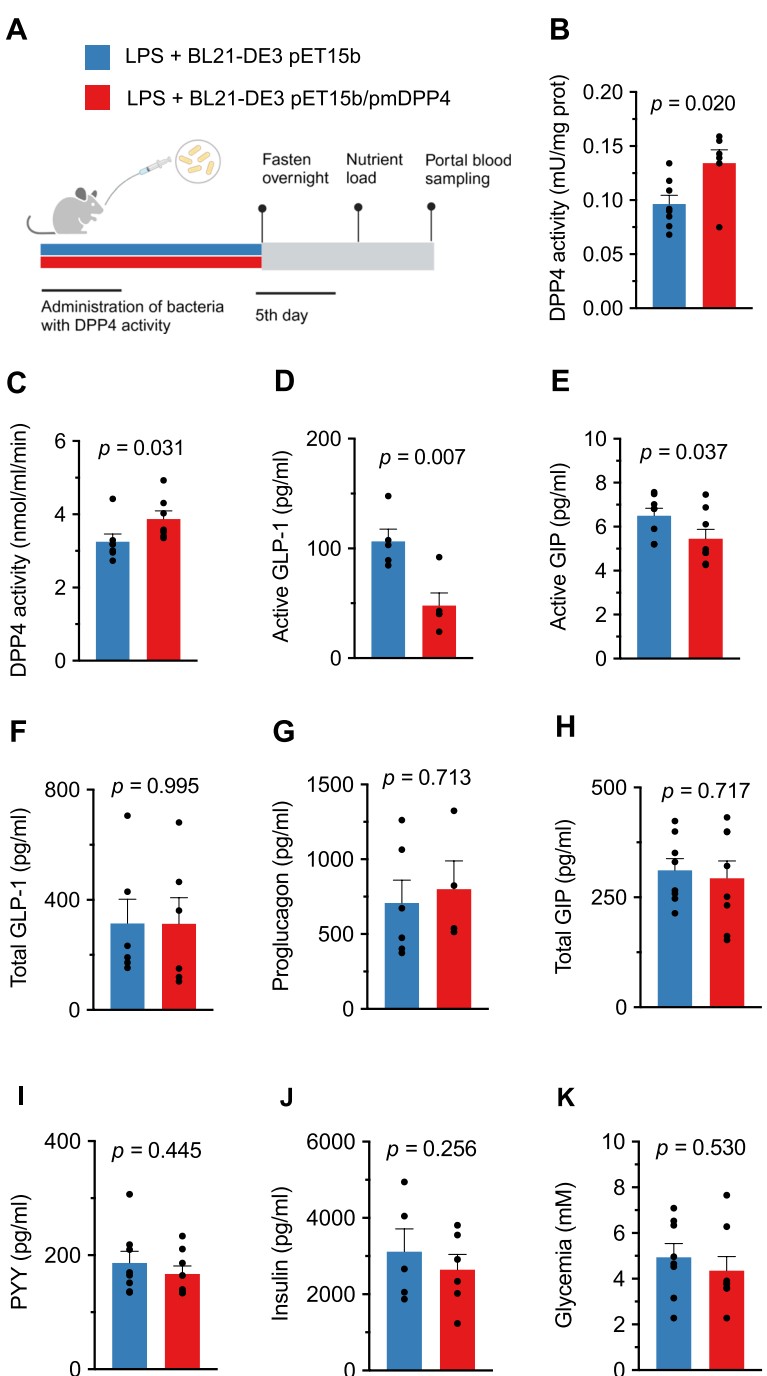

**Fig. 4** In vivo function of the pmDPP4 protein. Scheme of the animal study to test in vivo activity of bacterial DPP4-like enzyme. **A** Study design, mice administered with *E. coli* BL21-DE3 cells carrying the pET15b empty vector or the pET15b/pmDPP4 vector together with intraperitoneal injections of LPS (*n* = 8 each). **B, C** DPP4 activity measured in cecal content (**B**) and portal vein (**C**) of mice. In the portal vein, active GLP-1, active GIP, total GLP-1, proglucagon, total GIP, and PYY are shown in panels **D**, **E**, **F**, **G**, **H,** and **I**, respectively. Insulinemia and glycemia is shown in panels **J** and **K**. Data were analyzed using a *t*-test and significance was set at *p* < 0.05

GLP-1 and GIP (Fig. 4D,E). We excluded the existence of differences in hormone secretion, as no changes were found in total GLP-1, its precursor proglucagon or total GIP (Fig. 4F–H). There were also no differences in the total form of other hormones, such as PYY, or in insulinemia, or glycemia (Fig. 4I–K).

### P. merdae DPP4-like enzyme is differentially inhibited by gliptins

Finally, we evaluated the inhibition of bacterial DPP4-like and hsDPP4 proteins by currently approved gliptins: sitagliptin, vildagliptin, saxagliptin, and linagliptin. Lineweaver–Burk double-reciprocal plots of the kinetics showed that all inhibitors showed classical competitive inhibition of hsDPP4, affecting the $K_m$ but not the $V_{max}$ (Additional file 3: Figure S3). Overall, pmDPP4$\Delta$23 had an estimated $K_m$ of 1.63 mM, representing on average a 3.97 fold-change ($p < 0.001$) over the observed hydrolysis rate for hsDPP4 ($Km = 0.33$ mM) (Additional file 4: Table S1). As expected, the enzymatic activity hsDPP4 was inhibited by all gliptins, with sitagliptin, saxagliptin, and linagliptin having a more significant effect on its dipeptide hydrolysis function (Additional file 4: Table S1). Contrastingly, bacterial pmDPP4$\Delta$23 was only inhibited by saxagliptin and vildagliptin, which reduced its hydrolytic capacity on gpPNA (fold-change = 1.73, $p = 0.036$ and fold-change = 0.78, $p = 0.076$, respectively, at the highest inhibitor concentration) (Additional file 4: Table S1). The determination of $EC_{50}$ values confirmed the poor effect of gliptins on pmDPP4 function compared with hsDPP4 (Fig. 5). The $EC_{50}$ for all gliptins against hsDPP4 was in the nanomolar range ($< 10^{-6}$) with linagliptin showing the strongest inhibitory activity ($3.68 \times 10^{-9}$). Conversely, pmDPP4$\Delta$23 was only inhibited by vildagliptin and saxagliptin, but always to a lesser extent than hsDPP4 (twofold and 2.4-fold less, respectively) (Table 1). Non-linear fitting of the dose–response data showed an almost null effect of sitagliptin and linagliptin on pmDPP4$\Delta$23, leaving its hydrolytic activity intact (Fig. 5).

### Discussion

DPP4 activity, present on the surface of mammalian cells and as a soluble form in plasma, is crucial in regulating host metabolism by cleaving and deactivating the incretins that signal insulin release. The present study makes an advance by showing that the prevalence of bacterial DPP4-like genes correlates with human metabolic status in several metagenomic analyses. In addition, our in vitro and in vivo studies with the DPP4 of *P. merdae* provide a mechanistic rationale for the findings of our metagenomic assessment and argue for the detrimental impact of bacterial DPP4 on human health.

To best characterize the gut microbiota as a source of DPP4-like activity, we first searched for one of the millions of genetic traits encoded in the human gut microbiome based on its distant similarity to the human *DPP4* or *CD26* gene product. We show that human DPP4 has distant sequence-based homologs that are widely distributed in the genome of hundreds of Gram-negative bacterial species, Bacteroidota phylum members, some of which are prevalent members of the human gut microbiota. We demonstrate that the products of such bacterial genes are more similar to eukaryote DPP4 proteins (e.g., human, rat, and mouse) than to the bacterial XaaPro/PepX peptidases (so-called S15 family). The PepX family comprises a set of proteins often referred to as DPP4 enzymes, which has led to confusion in distinguishing the true DPP4-type enzymes

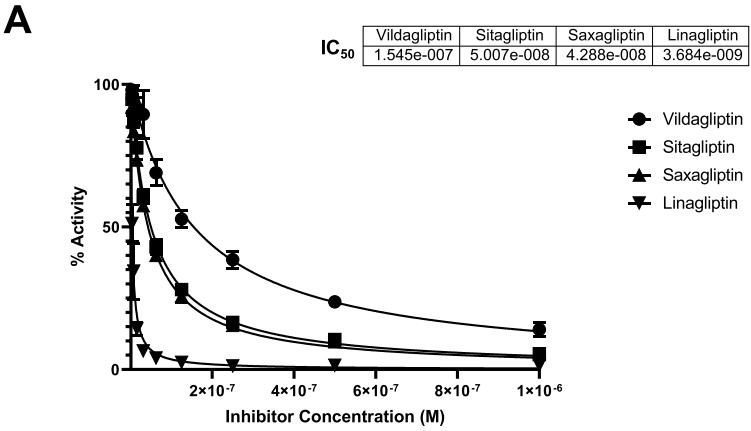

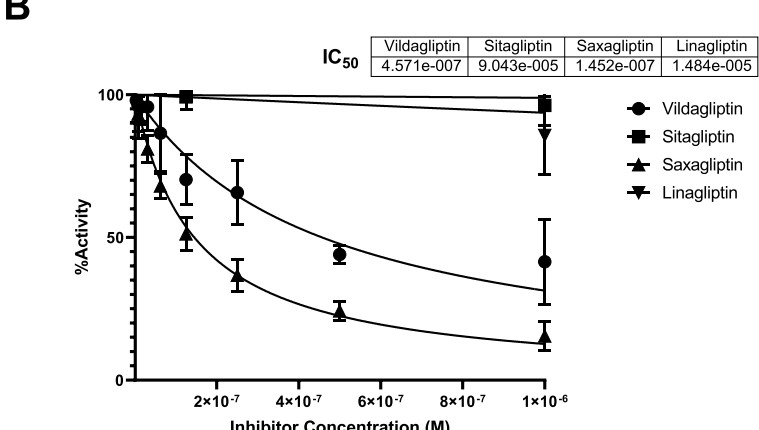

**Fig. 5** Dose–response of gliptins on DPP4 enzymes. The $EC_{50}$ parameter was calculated for hsDPP4 and pmDPP4Δ23 to determine the inhibitory efficacy of the drug on the hydrolytic activity of the enzymes. Serial dilutions of FDA- or EMA-approved gliptins from $1 \times 10^{-6}$ to $1 \times 10^{-9}$ M were established to measure the production of para-nitroaniline. Top panel (**A**) shows dose–response curves for hsDPP4, and bottom panel (**B**) shows dose–response curves for bacterial pmDPP4Δ23. Data obtained were computed via non-linear fitting regression to estimate the $EC_{50}$. Data were derived from four independent replicates using different protein purification batches

**Table 1** Estimation of $EC_{50}$ in a gliptins dose–response assay

| Gliptin | hsDPP4 | | | pmDPP4 | | |
|---|---|---|---|---|---|---|
| | $EC_{50}$ | CI | $R^2$ | $EC_{50}$ | IC | $R^2$ |
| Sitagliptin | $5.01 \times 10^{-8}$ | $4.76 \times 10^{-8}$ to $5.27 \times 10^{-8}$ | 0.995 | $9.04 \times 10^{-5}$ | $1.35 \times 10^{-5}$ to 1 | ND |
| Vildaliptin | $1.54 \times 10^{-7}$ | $1.41 \times 10^{-7}$ to $1.70 \times 10^{-7}$ | 0.980 | $4.57 \times 10^{-7}$ | $3.68 \times 10^{-7}$ to $5.72 \times 10^{-7}$ | 0.833 |
| Saxagliptin | $4.29 \times 10^{-8}$ | $4.14 \times 10^{-8}$ to $4.44 \times 10^{-8}$ | 0.998 | $1.45 \times 10^{-7}$ | $1.30 \times 10^{-7}$ to $1.69 \times 10^{-7}$ | 0.971 |
| Linagliptin | $3.68 \times 10^{-9}$ | $3.30 \times 10^{-9}$ to $4.10 \times 10^{-9}$ | 0.982 | $1.48 \times 10^{-5}$ | $4.87 \times 10^{-6}$ to 1 | ND |

$EC_{50}$ Estimated at molar (M) concentration, *CI* confidence interval at 95%, $R^2$ calculated as the fit of data into the non-linear function of the inhibitor concentration vs normalized response. Data obtained from independent experiments ($N = 2$). ND, data did not fit as non-linear function

probably because of their similar activity to trim X-A or X-P N-terminal dipeptides in many substrates [21, 22]. In the latter respect, the conserved and consensus GWSPGGF motif (GWSYGGY in eukaryotes) present in the bacterial DPP4 homolog has led to the

assignment of this peptidase to the S9B family of serine peptidases according to the MEROPS database [22]. Our results are in agreement with recent findings showing the presence of DPP4-like enzymes in several species of *Bacteroides* [14, 16].

We have also addressed one of the criticisms we received when our hypothesis paper was published, which questioned the ability of bacteria to secrete DPP4 [30]. In this sense, our new data suggest that bacterial DPP4 proteins contain N-terminal signal peptides that potentially drive secretion and are indeed similar to the transmembrane domains present in the hsDPP4 amino acid sequence. Using various cell-based assays and microscopy approaches, we show that the leader sequence of bacterial DPP4 proteins, exemplified by the 23 amino acid N-terminal peptide from *P. merdae*, serves as a signal for secretion. We conclude that, similar to hsDPP4, the bacterial DPP4-like protein tends to be located extracellularly rather than intracellularly as for PepX peptidases [31, 32]. In addition, bacterial DPP4-like proteins share other features with the human enzyme, such as similarities in the amino acid sequence (∼30%), domain architecture, quaternary structure, and catalytic motifs [29, 33].

Given the known role of hsDPP4 as a regulator of glucose homeostasis, we questioned the value of DPP4 expression in the gut microbiome. Our metagenomic assessment suggests that DPP4-like genes from *Parabacteroides* and *Porphyromonas* species may play an important role in T2D, as they are enriched in patients with T2D. Our results are in accord with the taxonomic evaluation performed in one of the largest cohorts of individuals with T2D published to date, where *Parabacteroides* appears as one of the genera enriched in T2D subjects [19]. Much less is known about the effect of *Porphyromonas* on host metabolism, likely because its predominant habitat is the oral cavity. Nevertheless, some studies have provided information on the metabolic deterioration caused by *Porphyromonas gingivalis*, with recently reported DPP4-like activity [16], when administered to mice [34]. We hypothesize that the mechanism underlying our results, as well as those reported by others, lies in the functional overlap of DPP4 activity within eukaryotic cells and gut microbes. Indeed, we show in vitro that the *P. merdae* DPP4-like enzyme inactivates GLP-1 and GIP as efficiently as hsDPP4. In mice sensitized by LPS, we confirm that the administration of *E. coli* overexpressing the full-length DPP4-like gene from *P. merdae* reduces the active forms of GLP-1 and GIP, which is associated with higher DPP4 activity in portal blood. The fact that we found no changes in blood glucose and insulin levels could be attributed to the lag between the release of incretins and their effect on glucose metabolism. Very recently, Wang et al. reported similar results regarding the potential of the DPP4-like activity encoded by *Bacteroides thetaiotaomicron, Bacteroides fragilis*, and *Bacteroides vulgatus* to inactivate GLP-1 [16]. Notably, both studies agree that the gut microbiota contributes to DPP4 activity and the decrease of active GLP-1 when the gut barrier is disrupted, either by the oral administration of a high-fat diet or detergents [16] or by intraperitoneal injections of LPS. We did not evaluate the effect of *P. merdae* DPP4-like enzymes in mice not treated with LPS, as we would expect no differences in the cleavage of incretins as Wang et al. described in mice fed a control diet. In addition to the inactivation of incretins, we also provide evidence for the ability of bacterial DPP4-like to process other gut peptides in vitro, including PYY and NPY, which would have further

implications for host behavior and mental health. For example, full-length PYY binds with similar affinity to all Y receptors, but the predominant circulating form of PYY-immunoreactivity, truncated $PYY_{3-36}$, preferentially binds to the Y2 receptor [35], a G protein-coupled receptor implicated in appetite and feeding control, mood, and mental disorders [36]. By degrading $NPY_{1-36}$ to $NPY_{3-36}$, DPP4 increases the affinity of NPY for Y2 and Y5 receptors but reduces its activity at Y1 receptors, which are known to be responsible for the antidepressant effects of NPY [37]. Accordingly, in mice fed an obesogenic diet, increases in DPP4 activity in systemic circulation have been associated with a depression-like behavior [38], an effect that is counteracted by sitagliptin [39]. Unfortunately, the lack of commercial kits to measure the cleaved forms of PYY and NPY makes their in vivo modification by the microbiota a matter for future studies.

The existence of gut bacteria with DPP4-like activity rationalizes the evaluation of their interaction with antidiabetic drugs with inhibitory DPP4 activity. According to our data, gliptins interact with DPP4 enzymes through competitive inhibition, indicating all gliptins occupy the substrate site of this enzyme. Sitagliptin and vildagliptin were the first drugs to be approved and are the most widely prescribed [8]. Our results indicate that DPP4 from *P. merdae* was minimally inhibited by sitagliptin (and linagliptin), in line with previous studies on the DPP4-like enzyme from *B. thetaiotaomicron* [16]. Thus, *P. merade* DPP4-like enzyme seems to have modified the substrate binding site enough to hindering non-peptidemimetic gliptin (e.g., sitagliptin and linagliptin) binding. By contrast, vildagliptin and saxagliptin—both with peptide-like chemistry [40]—have an effect on the DPP4 activity of *P. merdae*, with $EC_{50}$ values in the nanomolar range, suggesting other and novel gliptins with same chemistry would inhibit bacterial DPP4-like activity as well. For vildagliptin, concordant results in mice showed that the drug abolished DPP4 activity in the contents of the cecum and feces [41]. However, our results contrast with the recently characterized DPP4-like enzyme from *B. thetaiotaomicron*, which is inhibited in the nanomolar range ($10^{-8}$ range) by saxagliptin and in the micromolar range by sitagliptin ($10^{-6}$ range) and linagliptin ($10^{-5}$ range) [14]. These different inhibition patterns could be explained by the extensive sequence diversity of bacterial DPP4-like enzymes, resulting in a large repertoire of proteins with differing abilities to bind gliptins and affect their catalytic activity. Therefore, characterizing the function of DPP4-like enzymes from different gut bacterial species is crucial to elucidate their impact on T2D progression and gliptin therapy. Indeed, the information obtained from our inhibition studies could have significant consequences for the pharmacology of T2D treatment. For instance, the partial inhibition of the *P. merdae* DPP4-like activity by vildagliptin and saxagliptin suggests that bacterial enzymes can bind DPP4 inhibitors regardless of the inhibition efficacy. Given the substantial prevalence of DPP4-like enzymes in the intestinal lumen, this phenomenon could significantly affect the bioavailability of drugs targeting the human enzyme [42]. Furthermore, we and others have demonstrated that sitagliptin does not inhibit bacterial DPP4, highlighting the limitation of this drug to prevent incretin inactivation from all possible enzymatic sources. This observation likely explains the differential response to sitagliptin recently reported in individuals with T2D [16].

## Conclusions

We show that the gut microbiome encodes a large number of DPP4-like functional homologs, the prevalence of which positively correlates with T2D status. The mechanism underlying this association might be functional mimicry of bacterial DPP4-like enzymes, as exemplified by the ability of *P. merdae* DPP4-like enzyme to inactivate incretins. Our results of the characterization of the inhibitory activity of gliptins on bacterial DPP4-like enzymes may serve as a starting point for redefining the efficacy of current DPP4 inhibitors, as we show that some DPP4 inhibitors (vildagliptin and saxagliptin) are compromised by nonspecific binding to bacterial enzymes and others, such as sitagliptin, by the null effect on bacterial DPP4-like enzymes.

## Material and methods

### Search strategy for bacterial DPP4 homologs in the gut microbiome

We searched for bacterial homologs of hsDPP4 in metagenome-assembled genomes from our previous study on people with overweight and the metabolic syndrome (Bio-Project PRJNB25727) [20]. We used a Blast-based approach against human (SwissProt id: P27487), mouse (P28843), and rat (P14740) DPP4 amino acid sequences. Thresholds were set to at least 85% coverage in length and > 25% sequence identity at the protein level.

### Phylogeny tree of bacterial DPP4 and X-prolyl dipeptidyl peptidase families

The amino acid profile (via *hmmer3*) of the aligned DPP4-like (S9 peptidase family) sequences detected in the metagenome of overweight subjects with metabolic syndrome was used as bait to compile the most complete set of similar proteins present in the Ref-Seq NCBI database through PSI-Blast [43]. In a similar manner, the PepX (S15 peptidase family) from *Lactobacillus acidophilus* NCFM (UniProt id Q5FJC6) was used to retrieve the most complete set of similar enzymes potentially present in bacterial genomes. Multiple PSI-Blast iterations were run to select aligned protein sequences with coverage of > 95% of query length and ≥ 25% amino acid identity. Redundancy was removed to retain sequences at 99% identity. Finally, we obtained 925 non-redundant amino acid sequences of bacterial DPP4 homologs and 671 sequences for the PepX family of peptidases. Human (hsDPP4), mouse (mmDPP4), and rat (rnDPP4) sequences were merged with those previously obtained from bacterial metagenes as well as with the protein P19577 from *Dichelobacter nodosus* (member of the S17 peptidase family), the outlier for phylogeny reconstruction. In total, we performed a multiple sequence alignment of more than of 1600 protein sequences, using an iterative process with refinement steps [44, 45]. Amino acid profiles and consensus sequences for both peptidase families were computed using the Hidden Markov Model approach and the *hmmer3* package [46, 47]. The best tree explaining the evolutionary relationships between the two types of peptidases was built using ProtTest v3.0 [48], by selecting the best of the 120 evolutionary models implemented, according to AIC. Tree topology was optimized by using the maximum likelihood method. The amino acid sequences derived from hsDPP4 and the consensus amino acid sequences obtained for S15 and S9 peptidase families using *hmmer3* were subjected to further analysis to depict protein architecture and functional domains

using the SMART server [49]. In a similar manner, the SignalP v4.1 server was used to predict the presence of signal peptides in the amino acid sequences (http://www.cbs.dtu.dk/services/SignalP-4.1/) [50], whereas the TMHMM server was used to predict the presence of transmembrane helices (http://www.cbs.dtu.dk/services/TMHMM/).

### Cloning, over-expression and purification of bacterial DPP4

The *P. merdae* DPP4-like gene (pmDPP4) was amplified from DNA obtained from the type strain *P. merdae* DSM19495[T] (DSMZ, Braunschweig, Germany), using the primers DPP4-PMER-F gata*ccatg*gtgaaacgattagggtttggtgcttta (with an NcoI restriction site, underlined) and DPP4-PMER-R gata*ggatcc*ttatta*gtgatggtgatggtgatg*caaattatcaaataaaaagttagacat (with a BamHI restriction site, underlined) and a $6 \times$ His tag (italics) to produce a C-terminal tagged protein. Additionally, an open reading frame (ORF) encoding an N-terminal $6 \times$ His-tagged pmDPP4 protein with no signal peptide (pmDPP4Δ23) was amplified from same type strain *P. merdae* DSM19495[T] DNA using the primers DPP4-PMER-F23 gataccatggga*catcaccatcaccatcac*agcaaacgtgtcgatttaaaagaaattaca and DPP4-PMER-R2 gata*ggatcc*ttacaaattatcaaataaaaagttagacat. The PCR products of (~2.5 kb) were purified and mixed, and 500 ng of DNA was digested with 5U NcoI and BamHI restriction enzymes (New England Biolabs, Ipswich, MA) overnight at 37 °C. Digested fragments were cloned into the BamHI/NcoI pre-digested pET15b plasmid using the Fast-Link™ DNA Ligation Kit (Epicentre, Madison, WI), and *E. coli* DH5α cells were transformed with the ligation products. Colonies were selected on LB agar with 100 μg/mL ampicillin (Sigma) and checked by PCR for the presence of the correct DNA inserts. The inserts were analyzed by Sanger sequencing to ensure in-frame position of the ORFs and tags. The plasmids encoding pmDPP4 and pmDPP4Δ23 ORFs were isolated from DH5α cells and transferred to *E. coli* BL21-DE cells by heat-shock transformation. To over-express the recombinant proteins, freshly transformed *E. coli* BL21-DE cells containing the respective plasmids were grown on NZY Auto-induction LB medium (NZYTech, Lisbon, Portugal) at 37 °C for 20 h. Then, bacterial cells were recovered by centrifugation at 5000 *g* for 25 min at 4 °C and washed and resuspended in 10 mL of TBS buffer pH 8.0. Cells were disrupted by a pre-incubation with 500 μg lysozyme (Sigma) at room temperature for 10 min, followed by 4 cycles of bead-beating (FastPrep-24™ 5G bead beating grinder; MP Biomedicals, Santa Ana, CA) of 30 s, using 0.1-mm glass beads. The supernatant was obtained by centrifugation of the cell lysate at 16,000 *g* for 20 min at 4 °C and was then incubated with 2 mL TBS pH 8.0 pre-equilibrated Talon® Metal Affinity resin during 2 h at 4 °C in a vertical rotating mixer (Takara Bio, Kusatsu, Japan). The resin was washed with 10 volumes of TBS pH 8.0 buffer containing 0.1% Triton X-100 and 10 mM imidazol (both from Sigma). Elution was performed with 10 resin volumes of TBS pH 8.0 containing 150 mM imidazol. Proteins were concentrated by filtration of the eluate in 15-mL Amicon® Ultra 50 k filters (Millipore, Burlington, MA). Subsequently, a concentration step was performed in 0.5-mL Amicon® Ultra 50 k filters, retaining a final volume of 30–50 μL. Purity and concentration of the recombinant proteins was assessed by SDS-PAGE and the Qubit Protein Assay Kit (Thermo Fisher Scientific, Waltham, MA). Gel filtration was used to recover the recombinant protein fraction according to its expected size on an AKTA Ettan LC FPLC System (Amersham

Pharmacy Biotech, Amersham, UK), with a Superdex 200 5/150GL column and Ca/Mg-free sterile PBS pH 7.4 (MP Biomedicals, Aurora, OH) as the mobile phase.

### Immunogold staining and *electron* microscopy

*E. coli* BL21-DE cells containing the pET15b/pmDPP4 or pET15b/pmDPP4Δ23 plasmids were cultured in LB medium and induced to express the recombinant proteins as described above. *E. coli* BL21-DE cells harboring the pET15b were equally treated and used as a negative control for immunogold staining. After the induction of protein expression induction, 0.5 mL of cells was centrifuged at 3000 $g$ and washed with 1 volume of PBS. Cells were pelleted and sent to the Electron Microscopy Core Facility at SCSIE (University of Valencia) for fixation, embedding, and immunogold staining using a $6 \times$ His monoclonal antibody (Takara Bio) diluted 1/5000 as the primary antibody, and 10 nM anti-mouse IgG-Gold antibody (Sigma) as the secondary antibody, diluted 1/20. Images were acquired on a JEM-1010 transmission electron microscope (JEOL, Tokyo, Japan) operating at 100 kV equipped with an AMT RX80 (8 Mpx) digital camera. Image analysis was performed at the SCSIE.

### Protein secretion assay

A colorimetric assay was developed to detect pmDPP4 secretion dependent on the presence of the 23 amino acid leader peptide. *E. coli* BL21-DE3 cells containing the plasmids pET15b/pmDPP4 or pET15b/pmDPP4Δ23 were grown overnight at 37 °C in LB medium with 100 μg/mL ampicillin (Sigma). Overnight cultures were 1:100 refreshed in 5 mL ampicillin-containing LB medium and grown for a further 3 h at 37 °C with shaking. For recombinant protein over-expression, we added 0.1 mM IPTG (isopropyl β d-1-thiogalactopyranoside) and 100 μg/mL ampicillin to cultures grown under the same condition for a further 3 h. After protein induction, 1 mL of culture was recovered in a 1.5-mL sterile microtube and cells were pelleted at 3000 $g$ for 10 min at 4 °C. Pellets were resuspended in 1 mL of cold PBS and incubated on ice for 5 min. From the suspension, 100 μL was removed to estimate the colony formation unit (CFU) count on LB-ampicillin plates. The remaining cell suspensions were again pelleted and finally resuspended in 200 μL of cold PBS. The secretion assay consisted of adding 450 μL PBS to 50 μL cellular suspension, mixing, and incubating at 37 °C with 600 rpm shaking. One hundred microliters of the suspension was collected in new 1.5-mL sterile microtubes at 0, 30, 60, and 90 min of incubation and were maintained on the ice up until the end of the collection. To obtain cell-free supernatants, all aliquots were centrifuged at 6000 $g$ for 10 min at 4 °C and supernatants were recovered in 1.5-mL microtubes and stored on ice until processing. The minimal content of viable cells (almost cell-free nature) of micro-volume supernatants was confirmed by spreading aliquots on LB-Amp plates. The cell-free supernatants were used to determine DPP4 activity secreted into the medium according to the protocol described below. Bacterial cell lysis was determined in cell-free supernatants using the LDH Cytotoxicity Assay Kit (Thermo Fisher Scientific, Waltham, MA). Both DPP4 and LDH activities obtained from cell-free supernatants were normalized to the CFU counts.

**Colorimetric measurement of DPP4 activity and inhibition assays**

The hydrolytic activity of the bacterial DPP4 enzyme was measured using Gly-Pro p-nitroanilide hydrochloride (gpPNA) (Sigma), as described for assessing hsDPP4 activity [41]. A range of substrate concentrations from 2.5 to 0.31 mM gpPNA in 50 mM Tris–HCl pH 8.3 was used for kinetic analysis. The standard curve was established with free para-nitroaniline (PNA) (Sigma) ranging from 2 to 0.031 mM in 50 mM Tris–HCl pH 8.3. Briefly, 20 μL of each cell-free supernatant was placed in triplicate in a flat 96-well plate. Two hundred microliters of the respective PNA solutions were plated in duplicate to obtain the standard curve. A total of 180 μL of substrate solution (gpPNA) was added to bacterial supernatants and the enzymatic activity was monitored for 30 min at 37 °C (380 nm) in a VICTOR2 fluorometer/spectrophotometer (Perkin Elmer, Waltham, MA). The enzymatic activity ([PNA generated (nM) / hydrolysis time (min)] / pmole enzyme) was quantified between 10 and 30 min of incubation time (linear kinetics). Three independent experiments with different purified protein batches were performed. The same protocol was followed for the murine samples (cecal content and plasma from the portal vein). In the case of the cecal content, the values of DPP4 activity were normalized by the amount of protein, quantified by the Bradford method (Bio-Rad, Hercules, CA).

The commercially available FDA-approved human DPP4 inhibitors sitagliptin, saxagliptin, linagliptin, and vildagliptin (all from Sigma) were diluted in PBS to a 2 μM working solution. The inhibitory activity of the four gliptins on hsDPP4 and pmDPP4Δ23 was assessed at the ratios 1:10, 1:50, and 1:100 purified protein:inhibitor. Four independent replicates with different purified protein batches were performed. Dose–response assays to determine the $EC_{50}$ were carried out using serial dilutions of the inhibitors from 1 μM to 3.91 nM, and the relative activity was calculated with respect to reactions lacking the inhibitor.

**Peptidase activity of DPP4-like enzymes on gut hormones**

The DPP4 substrates GLP-1 7–37 fragment, GIP, NPY and PYY (all from Sigma) were diluted to 2.5 μg/μL in Ca/Mg-free sterile PBS pH 7.4. Human DPP4 (Sigma) was used as a control in PBS at 50 ng/μL (0.585 μM). The in vitro peptidase activity was assayed in a 10-μL reaction containing 5 μg substrate (100–150 μM) and 10 nM of hsDPP4 or pmDPP4Δ23 recombinant proteins (molar ratio substrate:enzyme, 10,000:1), mixed on ice and incubated at 37 °C for 10 min with gentle orbital shaking (600 rpm). To test the activity of the pmDPP4 secreted protein, the reaction was the same but the purified recombinant protein was replaced with the same volume of *E. coli* BL21-DE cells carrying pET15b, pET15b/pmDPP4, or pET15b/pmDPP4Δ23 plasmids incubated with 0.1 mM IPTG for 3 h at 37 °C. The cells were washed and diluted in PBS ($\sim 5 \times 10^6$ cells per reaction). Inhibition assays were performed under similar conditions but in the presence of 500 nM of vildagliptin or sitagliptin. Hydrolysis reactions were inactivated by incubation at 90 °C for 10 min. For reactions containing cells, the cells were first collected by centrifugation and then heat-inactivated. The inactivated reactions were stored on ice until processed for MS.

**MALDI-TOF analysis**

The MS analysis was performed at the SCSIE proteomic service (University of Valencia). Briefly, for every in vitro reaction performed, 1 µL of each sample was spotted onto the MALDI plate and, after the droplets were air-dried at room temperature, 1 µL of matrix (10 mg/mL alpha-cyano matrix in 70% acetonitrile, 0.1% trifluoroacetic acid) was added and allowed to air-dry at room temperature. The resulting mixtures were analyzed on a 5800 MALDI TOF/TOF instrument (AB Sciex, Framingham, MA) in positive reflector and linear mode. The raw mass spectra were released in ASCII format and were analyzed using *mMass* v5.5 software (http://www.mmass.org/).

**Bacterial DPP4-like in T2D human metagenomes**

A total of 220 samples (available metadata information matched with downloaded sequencing metagenomic data) derived from one of the largest metagenome projects on T2D (BioProject PRJNA422434) were evaluated for the distribution of bacterial DPP4-like genes. For read mapping against the non-redundant DPP4-like peptide database ($N = 971$), we used the *usearch* v8.0.1623 algorithm [51] with the following parameters: *-usearch_local*, *-id 0.7*, *-strand both*, and *-top_hit_only*. Alignments were filtered to retain those with more than 66% read length. Differential abundance of DPP4-like genes was assessed using Generalized Linear Mixed Models (GLMM, *lme4::glmer* R function) with prior filtering for low abundant features (*CoDaSeq::codaSeq.filter*, *min.occurrence = 0.25*, *min.reads = 1*), execution of *zCompositions::cmultRepl* function supporting a Bayesian-Multiplicative replacement of zero counts, and centered log-ratio (clr) transformation of compositional microbial abundance data (*CoDaSeq::codaSeq.clr*). Changes in DPP4-like gene abundance were assessed individually using sex, age, and body mass index (BMI) as covariates in the GLMM model. Genus enrichment analysis on significantly abundant DPP4-like genes in subjects with T2D ($N = 135$) compared to non-T2D individuals ($N = 85$) (from same cohort) was evaluated using the *chi-square* test with Monte-Carlo simulation. Lastly, the DPP4-like gene abundance from T2D ($N = 135$) and no-T2D ($N = 85$) samples was compiled in an integrative analysis supported by sparse Partial Least Square-Discriminant Analysis (sPLS-DA) algorithms implemented in the *DIABLO* v6.10.8 package [52]. A tuned sPLS-DA model was built using fourfold cross-validation (limiting over-fitting) across multiple iterations (*nrepeat = 20*). Selection of a reliable model better explaining sample classification was based on their performance after cross-validation (AUROC $\geq 0.70$).

**Animal study**

Experiments were performed using 16 male C57BL/6 J mice (7 weeks of age; Charles River Laboratories, Écully, France). Mice were housed in individually ventilated cages (4 mice/cage) in a Biosafety level 2 laboratory with a 12-h light/dark cycle and temperature-controlled conditions ($23 \pm 2$ °C). Mice were acclimatized for 10 days and had ad libitum access to food and water. Mice were randomized to receive for five consecutive days *E. coli* BL21-DE cells carrying the empty expression plasmid pET15b ($3 \times 10^8$ UCF/day) ($N = 8$) or carrying pET15b/pmDPP4 ($3 \times 10^8$ UCF/day) ($N = 8$). Both groups were also treated with LPS from *E. coli* O111:B4 (0.1 mg/kg, dissolved in sterile PBS) to

induce intestinal permeability, as previously described [53]. Mice were fasted on day 5. Mice then received an oral load of Ensure Plus® (0.0124 kcal/g) to induce the release of gut hormones [54]. At 12 min, mice were anesthetized with isoflurane. At min 15, blood from the portal vein was sampled in EDTA-containing tubes with or without a DPP4 inhibitor (Millipore, Burlington, MA). The plasma was recollected after centrifugations at 12,000 *g* for 3 min. Mice were sacrificed by cervical dislocation to minimize potential confounding effects, and mice of the different experimental groups were interspersed. The cecal content was also collected and snap-frozen in liquid nitrogen. All samples were stored at −80 °C. In plasma samples containing the DPP4 inhibitor, the levels of active and total GLP-1 and total GIP, PYY, insulin, and proglucagon were measured using the Luminex™ Mouse Metabolic Hormone Expanded kit (Merck & Co., Inc. Kenilworth, NJ). Also, active GIP was measured using an ELISA kit (CrystalChem, Zaandam, The Netherlands). Glycemia postmortem was quantified using the Glucose Colorimetric Detection Kit (Thermo Fisher Scientific). DPP4 activity was measured in an aliquot of plasma without DPP4 inhibitor and in the cecal content and feces, as described above.

### Statistical analysis

A normality test (Shapiro–Wilk) was conducted on data to select the most appropriate statistical comparison test (parametric or non-parametric) between sample groups and treatments. The *t*-test or Wilcoxon-rank sum test was applied to compare the in vitro enzymatic activity of hsDPP4 and pmDPP4 and the biochemical and physiological data from the animal study. A GLMM with covariate controls (e.g., age, sex, and BMI) was used to evaluate the differential abundance of DPP4-like genes in patients with T2D. The sequencing project was also included as a covariate when multiple cohorts of metagenome samples were assessed simultaneously. The sPLS-DA was used to quantify the power of bacterial DPP4-like genes for the classification of T2D samples, using a four-fold cross-validation. All tests were applied in R v4.1.2. Plots were obtained using the *ggplot2* R package or GraphPad v9 (San Diego, CA).

### Supplementary Information

> Additional file 1: Figure S1. Comparison of DPP4-like function-associated peptidases. Figure S2. *Parabacteroides merdae* recombinant proteins and activity. Figure S3. Gliptin inhibition kinetics on *Parabacteroides merdae* DPP4-like protein. Table S1. Kinetics fold-change upon gliptin inhibition on pmDPP4 and hsDPP4 enzymes.
>
> Additional file 2: Review history.

#### Acknowledgements
The authors thank the Computational Biology & Data Science Unit from Principe Felipe Research Center (CIPF) for providing access to its high-performance computing cluster, co-funded by European Regional Development Funds (ERDF/FEDER). Finally, the authors thank the technical staff of the proteomics, microscopy, and animal facilities of SCSIE University of Valencia and, in particular, I. Noguera, for their assistance in protocols and procedures described for animal experimentation.

#### Review history
The review history is available as Additional file 2.

#### Authors' contributions
MO, YS, and AB-P intellectually conceived the study. PH-C, SC-B, and AB-P performed the molecular biology experiments and the in vitro experiments. AB-P performed the bioinformatics and metagenomic analysis. MO and RL-G performed the animal experiments and sample analysis. MO, YS, and AB-P drafted the manuscript. All authors read the final version. YS and AB-P are responsible for its content.

**Peer review information**

**Funding**

 The authors acknowledge the following funding bodies that provided support for this study: European Commission 7th Framework Program through the MyNewGut project (Grant agreement No. 613979) and the Spanish Ministry of Science and Innovation (MICINN) (grant PID2020-119536RB-I00) to YS; Institute of Health Carlos III (ISCIII) via CP19/00132 grant to AB-P with funds from the European Social Fund (ESF/FSE), CIPF Predoctoral Training Fellowships 2022 funded by JANSSEN-CILAG—supporting the contract of PH-C, CIPF core funding for the contract of SC-B; Early Career Grant from the Society for Endocrinology award to MO. Spanish Government MCIN/AEI for the Center of Excellence Accreditation Severo Ochoa (CEX2021-001189-S/MCIN/AEI /https://doi.org/10.13039/501100011033).

**Availability of data and materials**

The raw metagenomic data was obtained through accessing the following bioprojects publicly available via European Nucleotide Archive (ENA): PRJNA422434 [55] and PRJEB25727 [56].

## Declarations

**Ethics approval and consent to participate**

The animal experiment was performed following European Union 2010/63/UE and Spanish RD53/201 guidelines, approved by the ethics committee of the University of Valencia (Animal Production Section, SCSIE, University of Valencia), and authorized (Ref 2022/VSC/PEA/0244) by the competent authority (Generalitat Valenciana).

**Competing interests**

The authors declare that they have no competing interests.

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

## 