## [Additional file 2: Review history. · Genome Biology]

Review History

First round of review

Reviewer 1

Are you able to assess all statistics in the manuscript, including the appropriateness of statistical tests used? No, I do not feel adequately qualified to assess the statistics.

Comments to author:

In their manuscript, the authors examine several DPP4 mimics found in the gut microbiota. They show that they maintain their specific activity when expressed heterologously in vitro and in vivo and that they respond to inhibitors. These genes are enriched in T2D patients' microbiomes. The mouse models show activity but not disease manifestation.

Major concerns:

This manuscript was preceded by papers showing that other bacterial homologs have activity (in particular, citations 14 and 16). These papers show both the activity and inhibition of these enzymes from *Bacteroides theta*, *Bacteroides vulgatus* and *Bacteroides fragilis*, relatives of *Parabacteroides merdae*. This significantly diminishes the novelty of the results. What is the protein sequence similarity between these proteins? How common are these genes across the *Bacteroidetes* genomes?

While BL21 *E. coli* is a practical choice for preliminary secretion studies of DPP4, its high secretion capacity and non-native status may not accurately represent in vivo conditions. Investigating whether *P. merdae* can naturally secrete these DPP4-like proteins is recommended. Should this be confirmed, using *P. merdae* in subsequent experiments would be advisable.

Similarly, the authors could have made a knockout of this gene in *P. merdae*. Is it essential? This is unclear from the wording in line 152.

The authors found that *E. coli* expressing DPP4 failed to induce T2D phenotypes (Figure 4J-K). A more biologically pertinent approach might involve an obese or T2D mouse model to better reflect the conditions associated with human metabolic disorders. Why didn't the authors use a T2D model to examine changes in these phenotypes?

The method used for cell removal involves a solitary centrifugation step, without subsequent validation to confirm the negligible presence of cells. To enhance the assay's robustness, implementing multiple centrifugation stages or the application of 0.22-micron filters is advisable. Additionally, culturing a sample of the cell-free supernatant could serve to verify the minimal bacterial presence.

Minor concerns:

Line 109 states their model is "better explained" compared to what?

There are many acronyms or methods that need to be spelled out or at least explained briefly in the text (WAG +I+G, GLMM, sPLS-DA, PYY, NPY etc.)

Correct the typo in the y-axis label of Figure 1, Panel D.

Amend typo on Line 384 to read "multiple sequence alignment of more than 1,600 protein sequences."

Correct the typo on Line 429 from "permanent mixing" to the intended term.

In discussing gene abundance data, it's challenging to separate the potential impact of gene abundance from that of species abundance. While the GLMM and sPLS-DA models reveal an association between certain species and T2D, attributing a causal relationship (whether complete or partial) necessitates further investigation. The role of DPP4 as a contributing factor is plausible, yet not definitively established, particularly as DPP4 genes from other species appear to exert dissimilar effects. Nevertheless, this is a known drawback of metagenomic analyses.

Reviewer 2

Are you able to assess all statistics in the manuscript, including the appropriateness of statistical tests used? Yes, and I have assessed the statistics in my report.

Comments to author:

The authors studied gut microbially encoded dipeptidyl peptidase 4 (DPP4)-like genes/enzymes in relationship to type 2 diabetes (T2D). In short, they find that several bacterial lineages harbour DPP4-like genes and such genes are more abundant in persons with T2D, based on publicly available metagenomics data. Importantly, they cloned and expressed the DPP4-like gene of *Parabacteroides merdae* and characterized its in vitro activity, showing that the enzyme is active against human incretins as well as neuropeptides. Notably, they also provide strong evidence that the enzyme is secreted, by using a truncated version of the cloned DPP4-like gene lacking leader sequences. Using mice treated with LPS, to induce gut barrier impairment, they showed that inoculation with *E. coli* expressing the *P. merdae* DPP4-like enzyme reduced active GIP and GLP-1 levels. Further, they show that compared to human DPP4 enzymes, common inhibitors are less effective, which has implications for treatment.

Overall, this is a well-written and carefully executed study that addresses an emerging topic (microbial mimicry) toward a mechanistic understanding of the role of the gut microbiome in health and disease -- the results should thus be of great interest to the readers. As such, I do not have any major concerns before acceptance/publication of the paper, and my comments/questions are mostly minor.

[1] For the in vivo experiments, it seems that a "leaky" gut may be important. Did the authors also test mice not treated with LPS, and presumably with an intact gut barrier (and thus less translocation of the enzyme)? Please discuss/clarify. Related, in the figure (Figure 4), it may be helpful to add the addition of LPS since this is probably a key experimental variable.

[2] It would be helpful to the reader if the authors could more clearly explain the differences between this work and the recent papers mentioned in the introduction; how is this study different and which specific question(s) that was not resolved in the other studies are being addressed? I understand that the authors recognize that their findings were partially "scooped" by the recently published study by Wang et al. (2023) but it would help the readers in situating this study.

[3] In the results section ("DPP4-like genes are widespread across human gut bacteria"), it would be helpful to briefly explain why PepX is being considered here. This is addressed in the Discussion section but it would be helpful to the readers to provide a brief explanation here too.

[4] For the metagenomics analysis, the authors used VSEARCH for mapping the sequencing reads. This should be ok but it seems that the identity threshold is quite lenient (70%), please clarify why not using common mapping methods such as bowtie2 or bwa. Also, please clarify how the mapping data (counts) were converted to abundance estimates (corrected for gene length, transcripts per million, or similar?). Although using compositionally aware techniques (zCompositions::cmultRepl and CoDaSeq::codaSeq.clr) is generally presumed to be less affected by variation in sequencing depth across samples (at least within a reasonable range), was variation in sequencing depth across samples accounted for (e.g., by rarefying)? Please clarify.

[5] Line 159. Please clarify the meaning of "Parabacteroides MULTISPECIES 99%".

[6] In the legend of Figure 1, it is mentioned that the taxonomy assignment was performed against the SILVA database. Unless I am misunderstanding, this is somewhat confusing since the SILVA is a 16S rRNA gene database, no? Also, the term "bait" is used (Line 975); I understand but would another term be more appropriate? Also, "abundance variation of bacterial DPP4-like proteins in human gut metagenomes" is mentioned; this should probably be DPP4-like protein-encoding genes (or similar), rather than DPP4-like proteins. Finally, should the tree include a scale bar to show evolutionary distances?

[7] Line 353: "A total of 220 samples (metadata available)". Please clarify the source of the metadata.

[8] The data availability mentions PRJEB25727 but this project is not mentioned in the text. Please clarify how/where these data were used.

Reviewer 3

Are you able to assess all statistics in the manuscript, including the appropriateness of statistical tests used? Yes, and I have assessed the statistics in my report.

Comments to author:

Summary

The authors identify functional homologues of the DPP4 enzyme within the gut microbiome, establishing a correlation between their gene abundance and patients with type 2 diabetes (T2D). Specific bacterial genera harbouring these genes include, but are not exclusive to, Bacteroides, Prevotella, Parabacteroides, and Porphyromonas - the latter two being independently and positively associated with T2D. In vitro and in vivo experiments demonstrate that, similar to the human DPP4 enzyme, these bacterial homologues are able to inactivate incretin hormones, and can be found extracellularly in the gut thanks to a leader peptide present in the full DPP4 homologue gene. Finally, the authors reveal that while current DPP4 inhibitor drugs (gliptins) effectively target human DPP4 enzymes, they exhibit reduced or negligible inhibitory effects against the bacterial DPP4-like homologues, opening an opportunity to improve T2D treatments.

Comments

- Generally speaking, the background could benefit from a small comment on why bacteria are producing these enzymes. What advantages do the enzymes have for the producers or the microbial community?

- It would be helpful to have context of how prevalent the bacterial enzymes (and associated gene sequences) are in T2D patients relative to baseline (non-T2D individuals). This may be pertinent to understanding T2D drug efficacy or if there will be responder and non-responder groups.

(i) How often are DPP4-like bacterial homologue enzymes found in T2D patients?

(ii) Could you please provide numbers of T2D patients vs non-T2D patients in terms of gene abundance?

- The authors refer to the gene abundance of DPP4-like sequences from different bacteria genera, attributing genera like Parabacteroides and Porphyromonas to T2D while Bacteroides and Prevotella were less abundant, but on the contrary, more present in non-T2D patients. Using the metagenomics data, could you represent such observation as a figure of showing the enrichment of those genera in the patients with and without T2D?

- The T2D study referred in the paper from where the enrichment of genes was taken comes from a Chinese cohort, due to geospecificity of the gut microbiome and variances in host genetics which modulate metabolism, is there another cohort to validate the presence of these genes/bacteria also in other T2D patients coming from other parts of the world?

- Figure 1A, it is not clear the distribution of genera across the strips. A colour legend could be added.

- Figure 1A - quality of tree would be ideally improved (higher resolution, more annotation). For example, the blue and red highlights branches is not very visible. Additionally, it is not very clear what the colour legend means.

- Within the same tree in Figure 1A, and as commented in line 130. The DPP4-like and PepX sequences were extracted from organisms not only coming from gut bacteria but also from environmental bacteria. Could the authors add the source information into the tree? To understand the origin of the peptidases and their similarity with the hsDPP4s. This can partially explain why drugs reduce hsDPP4 activity but not much pmDPP4. Figure 1B - please consider adding a title or similar to make it more clear the graph is comparing homology of genes versus bacterial sequences. Secondly, the layout is slightly confusing, please consider making the panelling which distinguishes the figures more clear.

- Could the authors comment on the mode of action of the Gliptin drugs in hsDPP4 enzymes and why their efficacy is reduced for pmDPP4?

- What do the authors mean in the material and methods section (Line 555) when saying "Selection of reliable models explaining variation was based on their performance (AUROC \geq 0.70)". What were these models used for?

- Could the presence/abundance of the species *Parabacteroides timonensis*, *Parabacteroides MULTISPECIES* and, *Porphyromonas catoniae*, without taking into account the DPP4 gene homologue, predict T2D status? (diagnostic? Responder/nonresponder)
- The authors could add the standard deviation on the AUROC plots (Figure 1D).
- Please add p-values to figure 4F-K to verify non-significance.
- Please consider removing tick marks (x-axis lines) on the Figure 4 bar plots
- Please add Panel letters from Figure 5 in its figure description (Top panel (A) and bottom panel (B)).
- Does *Bacteroides*-DPP4-like protein (non-T2D associated) have the same efficacy on incretins deactivation as the ones found in *Parabacteroides*?
- Line 384 in the methods section: "we performed a multiple sequence alignment of more than of >1,600 protein sequences, using an iterative process with refinement steps". In short, could you please provide details of the iterative process with refinement steps?

Point-by-point reply to reviewers

Reviewer #1: In their manuscript, the authors examine several DPP4 mimics found in the gut microbiota. They show that they maintain their specific activity when expressed heterologously in vitro and in vivo and that they respond to inhibitors. These genes are enriched in T2D patients' microbiomes. The mouse models show activity but not disease manifestation.

Major concerns:

This manuscript was preceded by papers showing that other bacterial homologs have activity (in particular, citations 14 and 16). These papers show both the activity and inhibition of these enzymes from *Bacteroides theta*, *Bacteroides vulgatus* and *Bacteroides fragilis*, relatives of *Parabacteroides merdae*. This significantly diminishes the novelty of the results. What is the protein sequence similarity between these proteins? How common are these genes across the *Bacteroidetes* genomes?

R/ The sequence identity between pmDPP4 and *Bacteroides* species orthologs is 53%, 54%, and 51% for *B. thetaiotaomicron*, *B. vulgatus*, and *B. fragilis*, respectively. Meanwhile, sequence identity is larger between *Bacteroides* species, reaching 82% in the case of *B. thetaiotaomicron* vs. *B. fragilis*. Therefore, we consider it plausible to think that the pmDPP4 protein has enough sequence divergence from *Bacteroides* orthologs to exhibit particular traits differential to those previously described for bacterial DPP4-like proteins (lines 181-182). On the other hand, we have previously described this type of proteins were exclusively found in the *Bacteroidota* phylum (formerly *Bacteroidetes*). We have reinforced this finding in the text (line 182-183).

While BL21 *E. coli* is a practical choice for preliminary secretion studies of DPP4, its high secretion capacity and non-native status may not accurately represent in vivo conditions. Investigating whether *P. merdae* can naturally secrete these DPP4-like proteins is recommended. Should this be confirmed, using *P. merdae* in subsequent experiments would be advisable.

R/ We politely disagree with the reviewer's argument. The BL21 strain was engineered and predominantly used to express soluble cytosolic proteins (doi: 10.1016/0022-2836(86)90385-2). Moreover, its use to overexpress membrane proteins, for instance, can cause stress by saturation of the translocation machinery (doi: 10.1074/mcp.M600431-MCP200). Its secretion capacity depends on specialized translocation machinery guided by the translocon complex (Sec, HlyA or T1SS/T3SS systems), which always responds to signal peptides encoded in the protein N-terminus (doi: 10.1002/elsc.201700200). Consequently, the acquisition of extracellular proteins must be further engineered to overcome this particular host's translocation control for recombinant protein production (doi: 10.1186/1475-2859-11-8). Besides, the experiments shown in Figure 2 and Supplementary material S2B demonstrate that the engineered protein's signal peptide guided exquisite secretion control, limited by secretory machinery availability in the host bacteria. While using the natural host is always desirable, the technical difficulties and long-term schedule of experiments for manipulating this particular strict anaerobe to tag the target native protein, construct single-gene mutants, and produce

monoclonal antibodies to trace secretion by TEM approaches followed, go beyond our study scope. Still, we thought the current configuration was valid enough to support our conclusions, and the massive genomic information compiled across hundreds of bacterial species was solid enough to describe the secretory nature of this family of proteins; an aspect neglected in the previous and recent studies.

Similarly, the authors could have made a knockout of this gene in *P. merdae*. Is it essential? This is unclear from the wording in line 152.

R/ The “essentially” adverb in the statement (now modified, lines 166-167) refers to the singularity rather than the indispensability nature of this genetic trait, which we could not address due to technical limitations. Nevertheless, the utilization of DPP4-like gene knock-out from *Bacteroides* species (doi: 10.1126/science.add5787) strongly suggests these genes are dispensable (lines 77-80).

The authors found that *E. coli* expressing DPP4 failed to induce T2D phenotypes (Figure 4J-K). A more biologically pertinent approach might involve an obese or T2D mouse model to better reflect the conditions associated with human metabolic disorders. Why didn't the authors use a T2D model to examine changes in these phenotypes?

R/ We aimed to demonstrate that DPP4 activity from the microbiota can inactivate incretins. We ruled out the use of mice with T2D because we hypothesize that DPP4 activity is a variable that alters glucose homeostasis and increases the risk of developing T2D. Thus, by using a model of T2D we could not address our hypothesis as such metabolic dysfunction is already present. We did consider using murine models of diet-induced obesity. However, in these, the obesogenic diet itself would have impaired the secretion of incretins (*Hira et al., 2020. doi: 10.1016/j.tem.2019.09.003*). We finally opted for an animal model in which LPS injections altered intestinal permeability (*Gou et al., 2013. doi: 10.1016/j.ajpath.2012.10.014*) and which, at the same time, resembles the state of metabolic endotoxemia well described in the context of obesity (*Cani et al., 2007. doi: 10.2337/db06-1491*). Before the sacrifice, we administered a nutrient load to induce incretin release.

We refer to Vannan et al.'s results (*Vannan et al., 2018. doi: 10.3389/fnut.2018.00089*), which studied the kinetics of glucose and gut hormone release in response to different nutritional challenges, including Ensure®. They described that peak GLP-1 and GIP release occur 15 minutes after Ensure® administration, but the peaks of glucose and insulin (measured by tail blood sampling) occur at 30 minutes. In our in vivo experiment, we administered Ensure®; 12 minutes later, we anesthetized the mice, and at minute 15, we took a blood sample from the portal vein. Animals were sacrificed, so we were unable to sample blood at a later time. We hypothesize that the absence of changes in blood glucose and insulin levels does not represent a failure of bacterial DPP4 to induce T2D phenotype. Instead, it responds to the lag between the release of incretins and their effect on glucose metabolism.

In view of the reviewer's comment, further details in this sense have been added in lines 322-324 and lines 599-600.

The method used for cell removal involves a solitary centrifugation step without subsequent validation to confirm the negligible presence of cells. To enhance the assay's robustness, implementing multiple centrifugation stages or the application of 0.22-micron filters is advisable. Additionally, culturing a sample of the cell-free supernatant could serve to verify the minimal bacterial presence.

R/ The micro-volumes obtained from such preparations make applying the filtering-based method for cell removal difficult. However, as the reviewer suggested, we have tested the cell-free supernatants obtained by centrifugation in plate culturing. We thought that the methodology used is valid for retrieving almost completely cell-free supernatants (see figure attached below), where viable and intact cells are minimal compared to the initial sample with full bacterial load. Even though viable cells were still present in supernatants (several magnitude orders below initial cell suspensions), we did not expect this to have a significant influence on the results obtained, as appropriate controls were included to conclude that the DPP4 activity measure is dependent on controlled protein secretion (see reply to concern#2) and not on cytosolic DPP4 proteins, which would require massive spontaneous cell lysis to alter the results. A statement in this regard was included in lines 502-502.

Minor concerns:

Line 109 states their model is "better explained" compared to what?

R/ Amended for better understanding (line 118).

There are many acronyms or methods that need to be spelled out or at least explained briefly in the text (WAG +I+G, GLMM, sPLS-DA, PYY, NPY etc.)

R/ Although most of them appear explained in the M&M section, we added more description over such acronyms (lines 119-120, 155, 171, 95).

Correct the typo in the y-axis label of Figure 1, Panel D.

R/ Amended.

Amend typo on Line 384 to read "multiple sequence alignment of more than 1,600 protein sequences."

R/ Amended, line 412.

Correct the typo on Line 429 from "permanent mixing" to the intended term.

R/ Amended, line 457.

In discussing gene abundance data, it's challenging to separate the potential impact of gene abundance from that of species abundance. While the GLMM and sPLS-DA models reveal an association between certain species and T2D, attributing a causal relationship (whether complete or partial) necessitates further investigation. The role of DPP4 as a contributing factor is plausible yet not definitively established, particularly as DPP4 genes from other species appear to exert dissimilar effects. Nevertheless, this is a known drawback of metagenomic analyses.

R/ Thanks to the reviewer for his/her feedback. Indeed, it is difficult to separate taxonomy from gene function in this context, and the DPP4 mimicry hypothesis could be explained by multiple microbiota-associated molecular mechanisms underlying T2D onset and development. That's why we stated across the text that further studies are needed to establish solid causal relationships in the future. Given the widespread distribution of such genes across gut microbial species, the role of functional redundancy should be further investigated. Nevertheless, with the apparently divergent functional capacities of the DPP4-like proteins and the modest sequence identity among them, we firmly think more precise links to T2D could be established in the near future.

Reviewer #2: The authors studied gut microbially encoded dipeptidyl peptidase 4 (DPP4)-like genes/enzymes in relationship to type 2 diabetes (T2D). In short, they find that several bacterial lineages harbour DPP4-like genes and such genes are more abundant in persons with T2D, based on publicly available metagenomics data. Importantly, they cloned and expressed the DPP4-like gene of *Parabacteroides merdae* and characterized its in vitro activity, showing that the enzyme is active against human incretins as well as neuropeptides. Notably, they also provide strong evidence that the enzyme is secreted, by using a truncated version of the cloned DPP4-like gene lacking leader sequences. Using mice treated with LPS, to induce gut barrier impairment, they showed that inoculation with *E. coli* expressing the *P. merdae* DPP4-like enzyme reduced active GIP and GLP-1 levels. Further, they show that compared to human DPP4 enzymes, common inhibitors are less effective, which has implications for treatment.

Overall, this is a well-written and carefully executed study that addresses an emerging topic (microbial mimicry) toward a mechanistic understanding of the role of the gut microbiome in health and disease -- the results should thus be of great interest to the readers. As such, I do not have any major concerns before acceptance/publication of the paper, and my comments/questions are mostly minor.

[1] For the in vivo experiments, it seems that a "leaky" gut may be important. Did the authors also test mice not treated with LPS, and presumably with an intact gut barrier (and thus less translocation of the enzyme)? Please discuss/clarify. Related, in the figure (Figure 4), it may be helpful to add the addition of LPS since this is probably a key experimental variable.

We did not evaluate the administration of *E. coli* carrying DPP4 activity in an animal model without LPS. We set up an animal model with an increased permeability to ease the bacterial DPP4 translocation, as we previously hypothesized that this factor would be fundamental (Olivares, et al., 2018. doi: 10.3389/fmicb.2018.01900). In accordance with our hypothesis, at the time of writing this manuscript, Wang et al. published that the DPP4 activity of *Bacteroides* only inactivates GLP-1 in animal models where intestinal permeability is increased, as occurs in obesity and colitis, but not in control mice.

This aspect has been clarified in the Discussion section lines 328-331. Additionally, "LPS" has been added to the legend of Figure 4, following the reviewer's suggestion.

[2] It would be helpful to the reader if the authors could more clearly explain the differences between this work and the recent papers mentioned in the introduction; how is this study different and which specific question(s) that was not resolved in the other studies are being addressed? I understand that the authors recognize that their findings were partially "scooped" by the recently published study by Wang et al. (2023), but it would help the readers in situating this study.

R/ Although we have already described in the introduction the findings that make this study go beyond the recently published, we have rewritten lines 89-98 to highlight our novel findings more explicitly.

[3] In the results section ("DPP4-like genes are widespread across human gut bacteria"), it would be helpful to briefly explain why PepX is being considered here. This is addressed in the

Discussion section but it would be helpful to the readers to provide a brief explanation here too.

R/ We have included a brief statement supporting the inclusion of this peptidase family in the analysis (lines 112-115).

[4] For the metagenomics analysis, the authors used VSEARCH for mapping the sequencing reads. This should be ok but it seems that the identity threshold is quite lenient (70%), please clarify why not using common mapping methods such as bowtie2 or bwa. Also, please clarify how the mapping data (counts) were converted to abundance estimates (corrected for gene length, transcripts per million, or similar?). Although using compositionally aware techniques (zCompositions::cmultRepl and CoDaSeq::codaSeq.clr) is generally presumed to be less affected by variation in sequencing depth across samples (at least within a reasonable range), was variation in sequencing depth across samples accounted for (e.g., by rarefying)? Please clarify.

R/ We did not use Bowtie2 or BWA algorithms because we completed the sequence level assessment analysis using our non-redundant DPP4-like protein database created in the first section of results. Bowtie2 and BWA algorithms are highly specialized for mapping short DNA reads against DNA references. USEARCH has the same sensitivity and specificity than Blast local alignment algorithm (ideal for aminoacids-based sequence comparison, based on small "word length") with faster execution. The identity threshold level (70%) is canonically used in protein-based comparisons and to maintain discrete protein family function and taxonomy assignment during the comparison, given the high divergence among DPP4-like proteins, even among close species (see reply to reviewer#1 first concern), and to maintain the framework used to survey the presence of distinctive DPP4-like proteins across overweight metagenomes (lines 393-394). We successfully used this same approach in previous metagenome-based analysis to map DNA short reads against protein databases (doi: 10.1002/mnfr.202000996, doi: 10.1128/mSystems.00209-19). On the other hand, DPP4-like proteins (and genes, by extension) have a quite similar length distribution (694-736 aa in length), which does not represent an issue during read count estimation. Finally, as reviewer#2 highlights, the compositional and sparse nature of microbiome data requires the application of appropriate methods to overcome differences in library size, as those employed in our pipeline described. Therefore, we did not rely on rarefying because of the risk of loss of sensitivity and statistical power to estimate differential abundance on DPP4-like genes.

[5] Line 159. Please clarify the meaning of "Parabacteroides MULTISPECIES 99%".

R/ Described as suggested in line 175-176.

[6] In the legend of Figure 1, it is mentioned that the taxonomy assignment was performed against the SILVA database. Unless I am misunderstanding, this is somewhat confusing since the SILVA is a 16S rRNA gene database, no? Also, the term "bait" is used (Line 975); I understand but would another term be more appropriate? Also, "abundance variation of bacterial DPP4-like proteins in human gut metagenomes" is mentioned; this should probably be DPP4-like protein-encoding genes (or similar), rather than DPP4-like proteins. Finally, should the tree include a scale bar to show evolutionary distances?

R/ We did use the SILVA database (the quasi-standard for bacterial taxonomy annotation) to group protein sequences at the family and genus levels. We used the species-level annotation retrieved from PSI-Blast to map and group sequences by family and genus across hierarchical categories of SILVA annotation to represent in the phylogenetic tree. The paragraph in the Figure 1 legend was corrected for better understanding (lines 829-832). The phylogenetic tree has been re-drawn, and now it contains genetic distances (internal grey circles).

[7] Line 353: "A total of 220 samples (metadata available)". Please clarify the source of the metadata.

R/ The study published by Qin et al. 2012 (doi: 10.1038/nature11450) assessed metagenomes from more than 350 Chinese individuals (including case and controls and validation cohort). We accessed the publicly available repository of metadata associated with such human samples (https://static-content.springer.com/esm/art%3A10.1038%2Fnature11450/MediaObjects/41586_2012_BFnature11450_MOESM372_ESM.xls). We then compare the metagenome datasets downloaded via European Nucleotide Archive server with available metadata information and we could match only 220 samples with metadata. Attempts to communicate with the authors of the original metagenome study in T2D subjects to obtain more matched information have been unsuccessful. We have further described this sentence for better understanding (lines 564-565).

[8] The data availability mentions PRJEB25727 but this project is not mentioned in the text. Please clarify how/where these data were used.

R/ Described in lines 393-394.

Reviewer #3: Summary

The authors identify functional homologues of the DPP4 enzyme within the gut microbiome, establishing a correlation between their gene abundance and patients with type 2 diabetes (T2D). Specific bacterial genera harbouring these genes include, but are not exclusive to, *Bacteroides*, *Prevotella*, *Parabacteroides*, and *Porphyromonas* - the latter two being independently and positively associated with T2D. In vitro and in vivo experiments demonstrate that, similar to the human DPP4 enzyme, these bacterial homologues are able to inactivate incretin hormones, and can be found extracellularly in the gut thanks to a leader peptide present in the full DPP4 homologue gene. Finally, the authors reveal that while current DPP4 inhibitor drugs (gliptins) effectively target human DPP4 enzymes, they exhibit reduced or negligible inhibitory effects against the bacterial DPP4-like homologues, opening an opportunity to improve T2D treatments.

Comments

- Generally speaking, the background could benefit from a small comment on why bacteria are producing these enzymes. What advantages do the enzymes have for the producers or the microbial community?

R/ A brief statement has been included in this regard in lines 77-80.

- It would be helpful to have context of how prevalent the bacterial enzymes (and associated gene sequences) are in T2D patients relative to baseline (non-T2D individuals). This may be pertinent to understanding T2D drug efficacy or if there will be responder and non-responder groups.

(i) How often are DPP4-like bacterial homologue enzymes found in T2D patients?

R/ We have explored the prevalence of DPP4-like genes from *Parabacteroides* species, and we found no drastic changes in occurrence between T2D subjects and controls, as those genes appear in almost all fecal metagenome samples (assuming presence as > 1 count). So, the impact would be related to the abundance, as demonstrated in our analysis.

(ii) Could you please provide numbers of T2D patients vs non-T2D patients in terms of gene abundance?

R/ Relative abundance is far from representing and reflecting an accurate measure of gene abundance distribution across samples for diagnostic aims, given interindividual variability and technical variation of different sequencing platforms. However, for reviewer#3 information aims, we calculated and measured such parameters from the T2D cohort analyzed. Globally, *Parabacteroides* DPP4-like genes have 1.3-fold more relative abundance in T2D subjects (0.112% vs 0.087%); with an extreme value of DPP4_927 gene, which accounts 1.98-fold more relative abundance in T2D than in controls (0.064% vs 0.033%). This last information was included in lines 162-165.

- The authors refer to the gene abundance of DPP4-like sequences from different bacteria genera, attributing genera like *Parabacteroides* and *Porphyromonas* to T2D while *Bacteroides*

and Prevotella were less abundant, but on the contrary, more present in non-T2D patients. Using the metagenomics data, could you represent such observation as a figure of showing the enrichment of those genera in the patients with and without T2D?

R/ As we understand from reviewer#2 request, the taxonomy analysis and association of bacterial genera abundance with T2D are outside the scope of our study as they were already explored in the original report (doi: 10.1038/nature11450), with the taxonomy associations described. We present a gene-centered approach to exploring a concrete function associated with the T2D condition and, to some extent, linked to certain bacterial species. Moreover, the relationship intuited by the reviewer (genera-to-T2D status) may not be retrieved through the proposed analysis as we explore and discriminate bacterial diversity at deeper taxonomy levels (species and strains).

- The T2D study referred in the paper from where the enrichment of genes was taken comes from a Chinese cohort, due to geospecificity of the gut microbiome and variances in host genetics which modulate metabolism, is there another cohort to validate the presence of these genes/bacteria also in other T2D patients coming from other parts of the world?

R/ This is an important consideration. Indeed, we tried to assess additional metagenome cohorts (which regularly have shorter sample sizes). Still, we faced technical obstacles, as some studies surveyed in PubMed have no deposited sequencing data, and the remaining ones containing deposited data in publicly accessed repositories are limited to providing the required metadata for sample classification and covariate control, even when we requested formally.

- Figure 1A, it is not clear the distribution of genera across the strips. A colour legend could be added.

R/ We have added a colour legend to external family tags.

- Figure 1A - quality of tree would be ideally improved (higher resolution, more annotation). For example, the blue and red highlights branches is not very visible. Additionally, it is not very clear what the colour legend means.

R/ The tree has been redrawn at a higher resolution (the maximum required for journal artwork is 600 dpi), making the outlier and human peptidase branches wider.

- Within the same tree in Figure 1A, and as commented in line 130. The DPP4-like and PepX sequences were extracted from organisms not only coming from gut bacteria but also from environmental bacteria. Could the authors add the source information into the tree? To understand the origin of the peptidases and their similarity with the hsDPP4s. This can partially explain why drugs reduce hsDPP4 activity but not much pmDPP4. Figure1B - please consider adding a title or similar to make it more clear the graph is comparing homology of genes versus bacterial sequences. Secondly, the layout is slightly confusing, please consider making the panelling which distinguishes the figures more clear.

R/ All commentaries and suggestions above regarding Figure 1's quality and design were incorporated into its new version. Besides, environmental DPP4-like genes are tightly linked to

the Flavobacteriaceae family. Therefore, such an annotation can be transferred. We have also added a title for panel B; the new position makes it independent from panel A.

- Could the authors comment on the mode of action of the Gliptin drugs in hsDPP4 enzymes and why their efficacy is reduced for pmDPP4?

R/ Our results indicate that all gliptins tested act on human DPP4 enzyme via competitive inhibition, binding directly to the substrate site and preventing its enzyme recognition and hydrolysis. The pmDPP4 activity is not inhibited by sitagliptin and linagliptin; curiously, both have non-peptidomimetic chemistry. Such evidence strongly suggests that peptide-like gliptins (e.g., vildagliptin and saxagliptin), as well as others with the same chemistry, would be more efficient in blocking bacterial DPP4-like activity. This requested comment has been briefly described in lines 347-357.

- What do the authors mean in the material and methods section (Line 555) when saying "Selection of reliable models explaining variation was based on their performance (AURQC 0.70)". What were these models used for?

R/ The sPLS-DA approach computes several models as the output of the classification evaluation. The models differ in the number of components (variables explored) the algorithm takes to improve the classification of samples. As a result, we can retrieve multiple classification models using one, two, or three variables (bacterial DPP4 genes, etc.). The filtering of those models to retain those that were more informative was based on their AUROC value obtained after cross-validation. We have slightly changed the sentence for better understanding (lines 584-585).

- Could the presence/abundance of the species *Parabacteroides timonensis*, *Parabacteroides MULTISPECIES* and, *Porphyromonas catoniae*, without taking into account the DPP4 gene homologue, predict T2D status? (diagnostic? Responder/nonresponder)

R/ As mentioned previously, the association of bacterial taxonomy with T2D is outside the scope of our study, as Qin and coworkers deeply assessed this matter in 2012 (doi: 10.1038/nature11450). They found butyrate producers enriched in controls and some *Bacteroides* and *Clostridium* species (as well as *Parabacteroides* species, see Table 1) enriched in T2D. For further insights into the T2D gut microbiome and clinical treatment response (metformin treatment), reviewer#2 should inspect the Forslund et al. 2015 study (doi: 10.1038/nature15766).

- The authors could add the standard deviation on the AUROC plots (Figure 1D).

R/ We have attempted to do so. However, the plots shown were produced by an internal function of the Bioconductor "mixOmics" library, which does not have the option or allow SD to be plotted across the ROC curve. Results from sPLS-DA and cross-validation are mixOmics-formatted. We have explored internally such a format, mixOmics R environment and function attributes and we saw no output associated with SD; therefore, it is difficult to export and read by other AUROC plotting alternatives.

- Please add p-values to figure 4F-K to verify non-significance.

R/ We have added the p-value to Figure 4F-K following the reviewer's suggestion.

- Please consider removing tick marks (x-axis lines) on the Figure 4 bar plots

R/ The tick marks have been removed.

- Please add Panel letters from Figure 5 in its figure description (Top panel (A) and bottom panel (B)).

R/ Corrected as suggested (line 890).

- Does Bacteroides-DPP4-like protein (non-T2D associated) have the same efficacy on incretins deactivation as the ones found in Parabacteroides?

R/ This is an intriguing question; we cannot describe any information in this regard as we're working on a new hypothesis addressing the functional heterogeneity of bacterial DPP4-like proteins. A comparative analysis must be done using proteins from different species and genera for such an aim. We expect to resolve such an inquiry in the near future.

- Line 384 in the methods section: "we performed a multiple sequence alignment of more than of >1,600 protein sequences, using an iterative process with refinement steps". In short, could you please provide details of the iterative process with refinement steps?

R/ The MUSCLE algorithm is widely recognized for its performance in multiple-sequence alignment (MSA) in comparative genomics, protein structure, and phylogenetic fields of investigation. The refinement step is based on sub-tree computing to establish a phylogenetic relationship among sequences. Considering prior phylogenetic relationships and scoring pairwise sequence alignments using common substitution matrices (e.g., BLOSUM62), the algorithm tries iteratively to get better scores by accommodating sequences in such sub-trees in different ways across every iteration. This information is detailed in the references cited for this statement (Edgar, RC. 2004a, Edgar, RC. 2004b).

Second round of review

Reviewer 1

I find the revisions acceptable.

Reviewer 3

The authors have properly answered to all my comments when possible.
I would only suggest to add the standard deviation as text in figure 1D if the authors were not able to show it in the image itself.